# Unleashing Guidance Without Classifiers for Human-Object Interaction Animation

**Ziyin Wang**[1]  **Sirui Xu**[1]  **Chuan Guo**[2]  **Bing Zhou**[2]  **Jiangshan Gong**[1]
**Jian Wang**[2]  **Yu-Xiong Wang**[1]  **Liang-Yan Gui**[1]

[1]University of Illinois Urbana-Champaign    [2]Snap Inc.

http://ziyinwang1.github.io/LIGHT

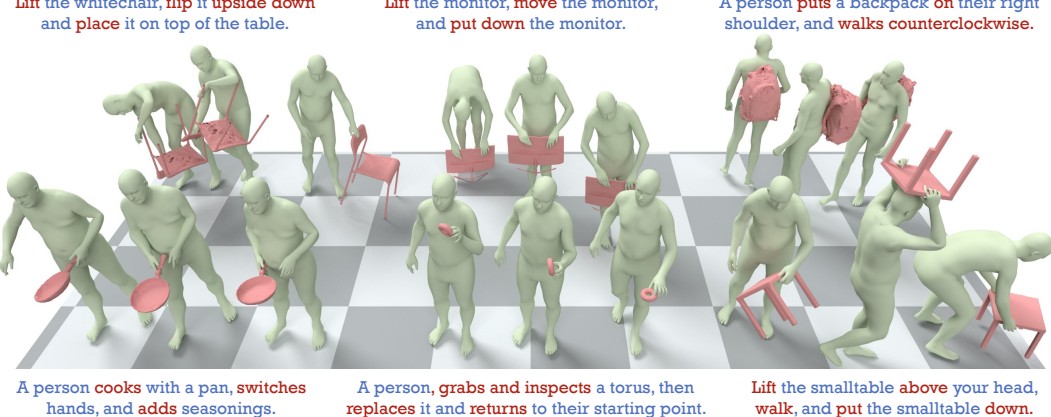

Figure 1: Given a textual prompt, our LIGHT generates realistic, vivid human-object interaction (HOI) motions via a novel classifier-free guidance scheme.

## Abstract

Generating realistic human-object interaction (HOI) animations remains challenging because it requires jointly modeling dynamic human actions and diverse object geometries. Prior diffusion-based approaches often rely on hand-crafted contact priors or human-imposed kinematic constraints to improve contact quality. We propose LIGHT, a data-driven alternative in which guidance emerges from the denoising pace itself, reducing dependence on manually designed priors. Building on diffusion forcing, we factor the representation into modality-specific components and assign individualized noise levels with asynchronous denoising schedules. In this paradigm, cleaner components guide noisier ones through cross-attention, yielding guidance without auxiliary classifiers. We find that this data-driven guidance is inherently contact-aware, and can be enhanced when training is augmented with a broad spectrum of synthetic object geometries, encouraging invariance of contact semantics to geometric diversity. Extensive experiments show that pace-induced guidance more effectively mirrors the benefits of contact priors than conventional classifier-free guidance, while achieving higher contact fidelity, more realistic HOI generation, and stronger generalization to unseen objects and tasks.

## 1 Introduction

Human-object interaction (HOI) animation, which involves generating dynamic motion sequences of a person interacting with objects, has become an important problem in computer vision and graphics. Realistic HOI animation underpins applications ranging from virtual reality, gaming, and robotics simulation, where agents must manipulate or use objects in a human-like way. Recent advances in generative model have opened new possibilities for synthesizing such animations directly from high-level specifications, *e.g.,* text descriptions. In particular, diffusion models (Peng et al., 2023;

Diller & Dai, 2024; Li et al., 2023a) have emerged as a powerful generative framework capable of modeling high-dimensional distribution of human motions and contacts, by iteratively denoising noise into plausible HOI sequences.

Despite this promise, a persistent challenge is ensuring high-quality realistic interactions between humans and objects. A plain diffusion model, without physics simulation, often produces artifacts: hands miss their targets, objects drift or penetrate the body, or contact is unstable over time. Prior work has sought to mitigate this through *guidance*. One line of methods employs external classifiers trained for *human-specified tasks*, *e.g.*, contact or affordance regression (Peng et al., 2023; Diller & Dai, 2024) to steer the diffusion process, but such classifier guidance is cumbersome to design, and may overfit to specific priors. Another line introduces *hand-crafted rules* with kinematic and dynamic constraints, such as forcing hands to align with objects via inverse kinematics (Xu et al., 2024), or introducing physics simulation (Xu et al., 2025b; 2026), but these approaches sacrifice generality and are computationally inefficient. Together, these efforts highlight: existing HOI animation still relies heavily on external priors not directly from the data.

Extending classifier-free guidance (CFG) (Ho & Salimans, 2022) to HOI animation is a natural way to reduce the reliance on external priors, but CFG, especially CFG based on text dropout for text-conditioned generation, mainly improves global distributional alignment and offers limited control over the fine-grained, persistent contact central to HOI. We therefore introduce LIGHT, *i.e., L*earning *I*mplicit *G*uidance for *H*uman-object in*T*eraction, a complementary data-driven framework where guidance arises from the relative denoising pace of separate components. As illustrated in Figure 1, LIGHT produces diverse and realistic human-object interaction animations with coherent contact dynamics. Concretely, we factor the representation into modality-specific components (*e.g.,* human and object) and assign each its own noise level with asynchronous schedules. And we formulate two paths: (**I**) a *staged schedule*, where one modality (*e.g.*, human) is kept cleaner while the other (*e.g.*, object) follows its prescribed noise levels, approximating a conditional trajectory; and (**II**) a *uniform schedule* that applies weak conditioning by assigning all modalities with prescribed noise levels, where the "unconditional" case is realized as noisier conditioning, constituting a *soft* form of CFG. The contrast between these two paths produces a guidance effect analogous to CFG. As the lag between schedules approaches zero, LIGHT collapses to joint denoising with no guidance; as the lag grows large, it approximates conditioning dropout in CFG. Our experiment shows that hard dropout on text improves global distributional alignment, whereas the soft guidance from LIGHT adjusts more on low-level contact details, indicating that LIGHT learns, purely from data, to reduce contact errors without hand-crafted priors. Note that assigning different noise levels to diffusion models naturally corresponds to the diffusion forcing mechanism (Chen et al., 2024), and LIGHT provides a principled extension of diffusion forcing into a guidance framework.

LIGHT relies solely on data priors; however, this does not constrain the model's flexibility to incorporate additional world knowledge like physics provided by humans, as such human-induced priors can also be reflected through data manipulation. We find that LIGHT achieves further improvement when training data are augmented with prior via *contact-aware shape-spectrum augmentation*. Specifically, objects are augmented with geometrically diverse alternatives from large repositories (Chang et al., 2015; Deitke et al., 2023), and motions are retargeted to preserve contact semantics and relative pose. These synthetic pairs teach the model that contact should remain invariant to irrelevant shape changes. This results in a stronger prior that the asynchronous schedules can exploit, improving generalizability to, *e.g.,* unseen objects from training.

In summary, our contribution lies in three folds:

- We introduce LIGHT, a novel guidance mechanism in which asynchronous denoising induces guidance without reliance on external classifiers. In contrast to CFG's reliance on hard dropout, our formulation provides a softer and more flexible alternative, naturally extending diffusion forcing into the guidance framework and potentially *inspiring future applications* in other domains.

- We propose contact-aware shape-spectrum augmentation, a strategy that preserves contact semantics while varying object geometry, thereby enabling the model to acquire more robust generative capabilities and improve generalization.

- We conduct extensive experimental evaluations demonstrating that the proposed approach consistently outperforms existing baselines and facilitates the generation of vivid and realistic interactions.

## 2 RELATED WORK

**Denoising Diffusion Models for Human Animation.** Denoising diffusion models (Sohl-Dickstein et al., 2015; Song et al., 2020; Ho et al., 2020) synthesize data by gradually denoising samples drawn from a noise distribution, effectively reversing a stochastic diffusion process. Recently, these models have achieved notable success in human motion generation, producing realistic and diverse animations (Barquero et al., 2023; Tevet et al., 2023; Zhang et al., 2022a; Raab et al., 2023; Zhang et al., 2023b; Shafir et al., 2023; Zhang et al., 2023e). A prominent example, MDM (Tevet et al., 2023), leverages transformer architectures to predict clean human motion trajectories from noisy inputs during the denoising stages. To generate conditional motion sequences, such diffusion methods typically incorporate external conditions into the diffusion process, including textual descriptions (Petrovich et al., 2023; Guo et al., 2022b; Petrovich et al., 2022; Zhang et al., 2022a; 2023d; Tevet et al., 2022a; Barquero et al., 2024). Further advancements have extended diffusion-based approaches to more conditioning scenarios, including guiding animations along specific trajectories (Karunratanakul et al., 2023; Rempe et al., 2023; Xie et al., 2023).

**Denoising Diffusion Models for Interaction Animation.** Recent advancements in interaction animation have shown significant progress in diverse scenarios, including human-human interactions (Liang et al., 2023; CMU; Mehta et al., 2018; Xu et al., 2023a;c) and human interactions within static scenes (Hassan et al., 2021; Cao et al., 2020; Hassan et al., 2019). Early human-object interaction (HOI) animation methods typically leveraged kinematic models combined with conventional generative frameworks. These earlier approaches primarily addressed simplified scenarios, either involving static or small objects (Xie et al., 2022; Zhang et al., 2020; Wang et al., 2022; Petrov et al., 2023; Taheri et al., 2022; Wu et al., 2022; Kulkarni et al., 2023; Zhang et al., 2022b), or focusing exclusively on hand-object interactions (Li et al., 2023c; Ye et al., 2023; Zheng et al., 2023; Zhou et al., 2022a; Zhang et al., 2024a; 2023a). Consequently, they struggled to effectively capture complex, dynamic, whole-body interactions. Recently, denoising diffusion models have emerged as a powerful paradigm capable of modeling sophisticated interactions involving dynamically moving objects (Peng et al., 2023; Diller & Dai, 2024; Li et al., 2023a; Wu et al., 2024a;b; Song et al., 2024; Xu et al., 2024; Zhang et al., 2024b). Concurrently, new datasets have expanded the range of possible interactions – from low-dynamic manipulations (Taheri et al., 2020) to highly dynamic scenarios engaging multiple body parts simultaneously (Bhatnagar et al., 2022; Jiang et al., 2023; Huang et al., 2022; Zhang et al., 2023c; Fan et al., 2023; Li et al., 2023b; Zhao et al., 2024; Kim et al., 2024b; Jiang et al., 2024; Yang et al., 2024). We use the most comprehensive InterAct dataset (Xu et al., 2025a) for evaluation.

**Guidance with Denoising Diffusion Models.** Diffusion models can be steered by additional guidance signals to better satisfy conditioning constraints. In image and text generation, for instance, classifier guidance (Dhariwal & Nichol, 2021) uses an external classifier's gradients to direct the denoising process toward desired outcomes, whereas classifier-free guidance (Ho & Salimans, 2022) foregoes a separate classifier by leveraging the model's own conditional and unconditional predictions for guidance. Inspired by such strategies, recent works in 3D human-object interaction (HOI) generation have explored explicit contact-aware or auxiliary-guided approaches to ensure realism. Without explicit constraints, diffusion-based HOI models often produce unrealistic artifacts, e.g., floating objects or missing contact points. To mitigate this, InterDiff (Xu et al., 2023b), for example, employs a kinematics-informed predictor that iteratively refines the diffusion output with corrections, improving human-object prediction accuracy. Similarly, HOI-Diff (Peng et al., 2023) integrates an auxiliary affordance module to steer the model toward consistent contact and affordance cues throughout generation. These approaches demonstrably boost realism but introduce additional complexity by relying on extra networks or hand-crafted constraints, which can hinder generalization. CG-HOI (Diller & Dai, 2024) cast interaction synthesis as a multi-task learning problem: an auxiliary contact prediction task is learned jointly to guide the motion generation, thereby avoiding training separate guidance models. However, it still imposes predefined relationships between predicted contact and object movements, using weighted combinations that embed human-designed assumptions. In contrast to these externally guided methods, we propose a guidance strategy without any assumptions on human-object contact, where the proposed approach gradually applies

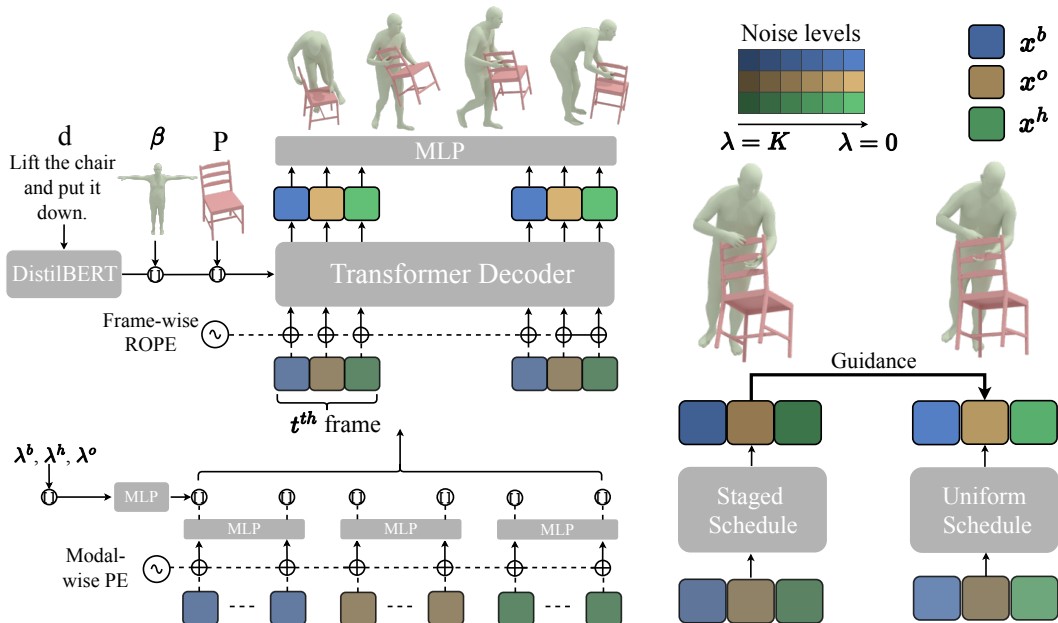

Figure 2: **Overview of LIGHT.** *Left: Training.* We form different modalities, *e.g.,* body, hand, and object, each diffused with its own noise level. After adding modal-wise and frame-wise rotary positional encodings, the tokens are processed by a shared Transformer decoder and an MLP head to predict clean motion. *Right: Inference.* We compare a *uniform* schedule that denoises all modalities synchronously with a *staged* schedule that keeps one modality cleaner from the uniform run.

guidance according to a temporal schedule for different components during denoising, promoting a new principled manner and and potentially inspiring future applications in other domains.

## 3 METHODOLOGY

**Overview.** We formalize the HOI animation task as the synthesis of a 3D motion sequence in which a human interacts with an object over a time horizon of $T$ frames. The input is a text description $d$, a canonical geometry of the object represented as a point cloud $P$, and a body shape parameter $\beta$ from SMPL-H (Loper et al., 2015; Romero et al., 2017), and the output of LIGHT is a sequence comprising both human motion and object motion trajectories. The human state is represented by joint positions $j^p \in \mathbb{R}^{T \times 52 \times 3}$, where 30 of total 52 joints are for both hands, and hand scalar rotation angles $j^{r_h} \in \mathbb{R}^{T \times 30}$. Object trajectories are in translations $o^t \in \mathbb{R}^{T \times 3}$ and rotations in a 6D representation (Zhou et al., 2019) $o^r \in \mathbb{R}^{T \times 6}$. Thus, each frame $x_t$ is fully characterized by the tuple $x_t = (j_t^p, j_t^{T_h}, o_t^t, o_t^r)$. We omit the time step $t$ in the following for simplicity.

**Preliminaries.** Diffusion forcing (Chen et al., 2024) is a recent advancement in diffusion modeling that generalizes the conventional diffusion process by allowing independent noise schedules for each token within a sequence. Unlike standard diffusion models, which uniformly apply a single noise schedule across all tokens at each diffusion step, diffusion forcing treats each token independently, enabling flexible and staged denoising processes tailored to sequential generation tasks. Formally, consider a sequence of tokens $x(0)$, representing the unperturbed data. Diffusion forcing introduces token-specific noise levels $\lambda$ with each element drawn independently from $\{0, 1, 2, \ldots, K\}$, where $K$ is the total number of denoising steps. Each token of $x(0)$ is then corrupted individually, $x(\lambda) = \langle \sqrt{\bar{\alpha}(\lambda)}, x(0) \rangle + \langle \sqrt{1 - \bar{\alpha}(\lambda)}, \epsilon \rangle$, where $\epsilon \sim \mathcal{N}(0, I)$ denotes Gaussian noise, and $\bar{\alpha}(\lambda)$ defines the cumulative noise schedule for handling tokens corrupted by varying degrees of noise. We use dot product $\langle , \rangle$ because the noise level for different tokens can be varied.

We slightly adjust the original diffusion forcing framework to directly predict the clean data $\tilde{x}(0) = \mathcal{G}_\theta(x(\lambda), \lambda, d)$ from its noised counterpart $x(\lambda)$, a typical solution for the motion generation task (Tevet et al., 2023), rather than predicting the noise. The training objective is formulated as minimizing the reconstruction error,

$$\mathcal{L}_{\text{DF}} = \mathbb{E}_{x(0), \lambda} \| \hat{x}(0) - \mathcal{G}_\theta(x(\lambda), \lambda, d) \|^2, \tag{1}$$

where $\hat{x}(0)$ denotes the ground truth data. $\mathcal{G}_\theta(x(\lambda), \lambda, d)$ denotes the model prediction of clean data from the generative model $\mathcal{G}$, parameterized by $\theta$, conditioned on the partially noised sequence $x(\lambda)$, the noise level $\lambda$ for each token, and the textual description $d$. For notational simplicity, we omit other inputs including human shape parameter $\beta$ and object geometry $P$ here; we describe how these additional conditioning variables are incorporated into our generative model below.

**Token Separation.** Following Cha et al. (2024), we explicitly decompose the representation $x$ into distinct modalities for the human body, hands, and objects, denoted as $x^b$, $x^h$, and $x^o$, respectively, which formulates these three components as separate token groups, yielding a total of $3 \times T$ tokens and noise levels $\lambda = \{\lambda^b, \lambda^h, \lambda^o\} \in \mathbb{R}^{T \times 3}$. As common practice for whole-body motion generation (Lu et al., 2023), this separation is beneficial for differentiating body and hand representations due to their distinct characteristics: the body encompasses 22 joints and typically exhibits larger-scale spatial movements, while the hands contain 30 joints with fine-grained and intricate finger motions. Empirically, we observe that separating the hand component improves motion generation quality in tasks with frequent hand interactions such as on the GRAB dataset (Taheri et al., 2020) (Figure 4).

**Overall Inference Procedure.** As shown in Figure 2, at test time, we perform two coupled denoising passes that share the same denoiser $\mathcal{G}_\theta$, as outlined in Algorithm 1. First, a *uniform* schedule denoises all modalities in sync under a same schedule, enhanced with text classifier-free guidance. Second, a *staged* schedule applies modality-specific schedules. The staged schedule incorporates the cleaner generation from the prior uniform schedule and introduces pace-induced guidance, combined with text CFG, as incremental guidance. The final sample is obtained from the staged schedule. Design choices and hyperparameters are provided in Sec. 4.

**Inference with Uniform Schedule.** The reverse process of diffusion forcing gradually denoises a noisy latent sequence $x_U(\lambda)$ into a clean sequence $x_U(0)$, defined as:

$$\tilde{x}_U = \mathcal{G}_\theta(x_U(\lambda), \lambda, d)$$
$$+ \omega_1\big(\mathcal{G}_\theta(x_U(\lambda), \lambda, d) - \mathcal{G}_\theta(x_U(\lambda), \lambda, \emptyset)\big), \tag{2}$$

$$x_U(\lambda - 1) = \langle\sqrt{\bar{\alpha}(\lambda - 1)}, \tilde{x}_U\rangle + \langle\sqrt{1 - \bar{\alpha}(\lambda - 1)}, \epsilon\rangle, \tag{3}$$

where our generative model predicts the denoised sequence $\tilde{x}_U$ given the partially noised input sequence $x_U(\lambda)$, with classifier-free guidance (CFG) with text $d$ dropout. $\omega_1$ is a hyperparameter to control the text CFG. With $\tilde{x}_U$ further noised back to $x_U(\lambda - 1)$, this iterative denoising step yields clearer HOI sequences until the inference advances to step $0$. In this scenario, all frames and modalities are denoised simultaneously as plain diffusion models, with $\lambda$ set to be the same and start at $K$. The resulting trajectory $x_U(\cdot)$ are incorporated into the staged schedule to formulate LIGHT.

**Inference with Staged Schedule.** Our *staged inference* strategy assigns asynchronized denoising schedules, enabling distinct denoising paces across modalities. Specifically, for specified modalities $m_1, m_2$ satisfying $m_1 \cap m_2 = \varnothing, m_1 \cup m_2 = \{b, h, o\}$. *i.e.,* together $m_1$ and $m_2$ cover all modalities without overlap (*e.g.,* $m_1 = \{h\}, m_2 = \{b, o\}$). This partitioning is flexible. In Table B, we verify that all possible combinations yield improvements to varying degrees. Then, we set

$$x'_S = \big(x_U^{m_1}(\lambda^{m_1} - \delta); x_S^{m_2}(\lambda^{m_2})\big), \quad \lambda' = \big((\lambda^{m_1} - \delta); \lambda^{m_2}\big), \tag{4}$$

where the vectors $\lambda^{m_1}$ and $\lambda^{m_2}$ are all-$k$ vectors at denoising step $k$, and $(\cdot\,;\,\cdot)$ denotes the concatenation operation. The offset vector $\delta$ determines how much earlier $m_1$ is denoised compared to $m_2$, thereby creating a pacing discrepancy across modalities. This discrepancy allows $m_1$ to reach a cleaner state from the previous generation $x_U^{m_1}$, and when it is combined with the current output of $x_S^{m_2}$ to formulate $x'_S$, it can lead to what we refer to as pace-induced guidance. Formally, the guided update is given by

$$\tilde{x}_S = \mathcal{G}_\theta(x_S(\lambda), \lambda, d)$$
$$+ \omega_1(\mathcal{G}_\theta(x_S(\lambda), \lambda, d) - \mathcal{G}_\theta(x_S(\lambda), \lambda, \varnothing)) \tag{5}$$
$$+ \omega_2(\mathcal{G}_\theta(x'_S, \lambda', d) - \mathcal{G}_\theta(x_S(\lambda), \lambda, d)), \tag{6}$$

$$x_S(\lambda - 1) = \langle\sqrt{\bar{\alpha}(\lambda - 1)}, \tilde{x}_S\rangle + \langle\sqrt{1 - \bar{\alpha}(\lambda - 1)}, \epsilon\rangle, \tag{7}$$

where $\omega_1, \omega_2$ are the scalar weights controlling the strength of the conditional influence. The resulting guided prediction for the $m_2$ component is then re-noised according to the schedule and propagated to the next denoising step, while $m_1$ continues to follow the trajectory of $x_U(k), k \in \{0, 1, 2, \ldots, K\}$.

---

**Algorithm 1** Inference with pace-induced guidance

---

**Require:** Total number of denoising steps $K$; lagged modality index $m_2$; cleaner modality index $m_1$; $\boldsymbol{x}_U(\boldsymbol{K})$ and $\boldsymbol{x}_S(\boldsymbol{K})$ initialized with same Gaussian noise; lag offset vector $\boldsymbol{\delta}$
1: **for** $k = K, K-1, \ldots, 1$ **do**
2:     Assign $\boldsymbol{\lambda}$ as an all-$k$ vector
3:     Obtain denoised $\tilde{\boldsymbol{x}}_U$ from $\boldsymbol{x}_U(\boldsymbol{\lambda})$ and $\boldsymbol{\lambda}$ with text CFG         ▷ Equation 2
4:     Add noise back to $\boldsymbol{x}_U(\boldsymbol{\lambda} - \mathbf{1})$ and record         ▷ Equation 3
5: **end for**
6: **for** $k = K, K-1, \ldots, 1$ **do**
7:     Assign $\boldsymbol{\lambda}$ as an all-$k$ vector
8:     **if** $\boldsymbol{\lambda} - \boldsymbol{\delta} \geq \mathbf{0}$ for all elements **then**
9:         Obtain the input of conditional branch $\boldsymbol{x}'_S(\boldsymbol{\lambda})$ by merging $\boldsymbol{x}_U(\boldsymbol{\lambda} - \boldsymbol{\delta})$     ▷ Equation 4
10:         Obtain denoised $\tilde{\boldsymbol{x}}_S$ from $\boldsymbol{x}_S(\boldsymbol{\lambda})$ and $\boldsymbol{\lambda}$ with text CFG     ▷ Equation 5
11:         Update $\tilde{\boldsymbol{x}}_S$ with $\boldsymbol{x}'_S(\boldsymbol{\lambda})$ by LIGHT       ▷ Equation 6, incremental guidance
12:     **else**
13:         Obtain denoised $\tilde{\boldsymbol{x}}_S$ from $\boldsymbol{x}_S(\boldsymbol{\lambda})$ and $\boldsymbol{\lambda}$ with text CFG
14:     **end if**
15:     Add noise back to $\boldsymbol{x}_S(\boldsymbol{\lambda} - \mathbf{1})$         ▷ Equation 7
16: **end for**
17: **return** $\boldsymbol{x}_S(\mathbf{0})$

---

**Architecture.** The left figure in Figure 2 illustrates the full architecture $\mathcal{G}_\theta$. (i) *Motion tokens.* At each frame we form three motion tokens, body $\boldsymbol{x}_t^b$, hand $\boldsymbol{x}_t^h$ and object $\boldsymbol{x}_t^o$, and attach a *modal-wise positional encoding* that distinguishes these modalities. A small linear projection is applied to each modality so that the three streams share a common hidden dimensionality before fusion. (ii) *Noise-level and temporal encodings.* Each motion token receives an additive noise-level embedding together that encodes the diffusion step with a *frame-wise timestep embedding*. (iii) *Object-geometry token.* Object shape is encoded once per sequence. From the point cloud $\boldsymbol{P}$, we concatenate an un-normalised Basis Point Set (BPS) (Prokudin et al., 2019) descriptor with a normalized BPS (Zhang et al., 2024c) whose maximal point-to-centroid distance is normalized to $0.95$; the original object scale (largest radius) is appended as an extra scalar. This geometry vector is processed by an MLP and then concatenated to text tokens. (iv) *Text token.* The input prompt $\boldsymbol{d}$ is encoded by a frozen DistilBert (Sanh et al., 2019) text encoder, producing a sequence of language tokens. These tokens are injected into each of the transformer decoder's layers through their cross-attention blocks. (v) *Transformer decoder and prediction head.* The concatenated sequence, including body, hand and object tokens with all positional, temporal embeddings, is passed through a Transformer decoder, with text and object-geometry tokens injected by cross-attention. A lightweight MLP head maps the denoised motion latents to final output, which are trained to match the clean ground-truth targets.

**Training Schedule.** Following Kim et al. (2024a); Xiu et al. (2025); Zhang et al. (2024d); Chen et al. (2024), we assign independent noise levels across modalities, formally defined as $\boldsymbol{\lambda}^b, \boldsymbol{\lambda}^h, \boldsymbol{\lambda}^o \sim U\{\mathbf{0}, \mathbf{1}, \mathbf{2}, \ldots, \boldsymbol{K}\}$ independently. Training with independent noise levels encourages the model to capture a broad range of asynchronized conditional distributions, enabling diverse conditionalities at inference. Unlike diffusion forcing (Chen et al., 2024), we do not vary noise levels across frames; instead, all frames within the same modality share the same noise level.

**Training Loss.** The model is trained with a composite loss consisting of a diffusion supervised term $L_{\text{DF}}$ and a regularization term $L_{\text{reg}}$. The total loss is defined as $\mathcal{L} = \mathcal{L}_{\text{DF}} + \mathcal{L}_{\text{reg}}$. The regularization term $L_{\text{reg}}$ promotes plausible human-object interactions and comprises three components: (i) a *bone-length loss* that penalizes deviations in limb lengths from ground truth, (ii) a *contact loss* that aligns designated human joints with expected contact regions on the object, and (iii) a *velocity loss* that matches predicted object and human motion velocity with ground truth. Full details of these components are provided in Sec. A.1 of the Appendix.

**Contact-Aware Shape-Spectrum Augmentation.** We employ an optimization-based augmentation strategy to enhance our dataset by transferring original HOI sequences onto novel objects from the same category. Specifically, we first train a correspondence network following Xie et al. (2024) that maps points from the source object's surface to corresponding points on new object surface selected from ShapeNet (Chang et al., 2015) and Objaverse (Deitke et al., 2023) within the same object category. With the learned correspondence, we replace the original object with the novel

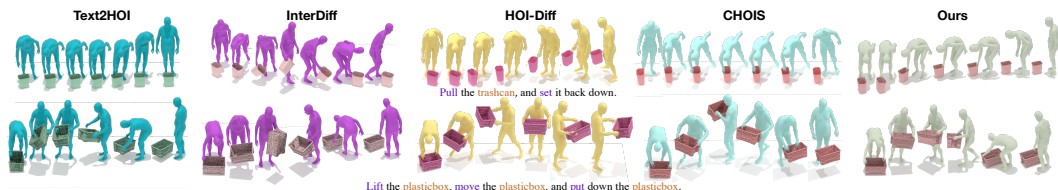

Figure 3: **Qualitative comparison** with baselines. Our method yields more realistic human-object interactions, fewer contact/penetration artifacts, more accurate finger positioning, and better text-motion alignment.

Table 1: **Quantitative comparisons** on the InterAct dataset (Xu et al., 2025a) between our method and baseline approaches. We report R-Precision with batch sizes 256.

| Method | R-Precision↑ | | | FID↓ | MM Dist↓ | Diversity→ | FSR↓ | Pene↓ | Contact→ | Interaction↑ | | |
|---|---|---|---|---|---|---|---|---|---|---|---|---|
| | Top 1 | Top 2 | Top 3 | | | | | | | $C_{prec}$ | $C_{rec}$ | $C_{F1}$ |
| Ground Truth | $0.600^{\pm0.004}$ | $0.834^{\pm0.001}$ | $0.909^{\pm0.001}$ | $0.000^{\pm0.000}$ | $1.475^{\pm0.003}$ | $7.781^{\pm0.140}$ | $0.083^{\pm0.204}$ | $0.076^{\pm0.000}$ | $0.208^{\pm0.000}$ | $1.000^{\pm0.000}$ | $1.000^{\pm0.000}$ | $1.000^{\pm0.000}$ |
| HOI-Diff (Peng et al., 2023) | $0.413^{\pm0.010}$ | $0.624^{\pm0.009}$ | $0.740^{\pm0.011}$ | $0.689^{\pm0.031}$ | $3.029^{\pm0.007}$ | $7.620^{\pm0.085}$ | $0.072^{\pm0.160}$ | $\mathbf{0.103}^{\pm0.010}$ | $0.084^{\pm0.006}$ | $0.722^{\pm0.002}$ | $0.447^{\pm0.026}$ | $0.501^{\pm0.022}$ |
| CHOIS (Li et al., 2023a) | $0.439^{\pm0.005}$ | $0.660^{\pm0.002}$ | $0.766^{\pm0.003}$ | $0.572^{\pm0.008}$ | $2.781^{\pm0.017}$ | $\mathbf{7.717}^{\pm0.115}$ | $0.119^{\pm0.234}$ | $0.131^{\pm0.013}$ | $0.115^{\pm0.002}$ | $0.710^{\pm0.012}$ | $0.520^{\pm0.004}$ | $0.541^{\pm0.007}$ |
| InterDiff (Xu et al., 2023b) | $\mathbf{0.501}^{\pm0.009}$ | $\mathbf{0.722}^{\pm0.003}$ | $\mathbf{0.824}^{\pm0.007}$ | $0.215^{\pm0.001}$ | $\mathbf{2.461}^{\pm0.013}$ | $\mathbf{7.717}^{\pm0.117}$ | $0.092^{\pm0.178}$ | $0.116^{\pm0.011}$ | $0.124^{\pm0.003}$ | $0.715^{\pm0.000}$ | $0.562^{\pm0.000}$ | $0.584^{\pm0.003}$ |
| Text2HOI (Cha et al., 2024) | $0.428^{\pm0.003}$ | $0.637^{\pm0.003}$ | $0.745^{\pm0.000}$ | $0.331^{\pm0.003}$ | $2.665^{\pm0.002}$ | $7.631^{\pm0.000}$ | $\mathbf{0.055}^{\pm0.123}$ | $0.105^{\pm0.003}$ | $0.102^{\pm0.001}$ | $0.711^{\pm0.001}$ | $0.488^{\pm0.000}$ | $0.532^{\pm0.002}$ |
| LIGHT (**Ours**) w/o guidance | $0.395^{\pm0.012}$ | $0.599^{\pm0.009}$ | $0.715^{\pm0.004}$ | $0.196^{\pm0.014}$ | $2.885^{\pm0.004}$ | $7.708^{\pm0.041}$ | $0.066^{\pm0.165}$ | $0.121^{\pm0.003}$ | $0.123^{\pm0.010}$ | $\mathbf{0.739}^{\pm0.008}$ | $0.567^{\pm0.019}$ | $0.599^{\pm0.018}$ |
| LIGHT (**Ours**) w/ guidance | $0.421^{\pm0.014}$ | $0.637^{\pm0.016}$ | $0.754^{\pm0.016}$ | $\mathbf{0.148}^{\pm0.014}$ | $2.756^{\pm0.018}$ | $7.712^{\pm0.050}$ | $0.078^{\pm0.197}$ | $0.132^{\pm0.001}$ | $\mathbf{0.132}^{\pm0.001}$ | $0.731^{\pm0.002}$ | $\mathbf{0.615}^{\pm0.016}$ | $\mathbf{0.627}^{\pm0.011}$ |

object, optimizing the placement so that the original human-object contact points are preserved – the new object's corresponding points remain consistently matched to the same human contacts. The optimization objectives are detailed in Sec. A.2 of the Appendix.

## 4 EXPERIMENTS

**Dataset.** We conduct experiments mainly on the InterAct dataset (Xu et al., 2025a), and ablate on its major subsets, BEHAVE (Bhatnagar et al., 2022) and OMOMO (Li et al., 2023a). InterAct includes fine-grained textual annotations accompanying these sequences, and we adopt its official training-testing split for all evaluations. As the data originally annotate human motion using SMPL-H (Romero et al., 2017) and SMPL-X (Pavlakos et al., 2019), we standardize all representations to SMPL-H and utilize SMPL-H joints consistently across our experiments, using official SMPL conversion. We manually filter out implausible motion sequences from the original MoCap data; for instance, we exclude cases from OMOMO in which hand orientations were incorrectly inverted, and cases from IMHD where the human is distorted given their shape and pose. We apply object augmentation to enrich the dataset from 217 objects to 1121 objects, with examples in Figure A.

**Metrics.** Following the standard practice (Guo et al., 2022a), we measure realism and diversity of the generated HOI sequence, alignment with textual descriptions, and physical plausibility of interactions. To evaluate realism and diversity, we utilize the Fréchet Inception Distance (**FID**), which quantifies feature distribution similarity between generated sequences and ground-truth samples, and a **Diversity** metric, measuring variability across generated HOIs. To assess textual alignment, we adopt **R-Precision**, and Multimodal Distance (**MM Dist**), quantifying the feature-level distance between generated HOIs and corresponding text embeddings. We introduce three metrics tailored explicitly for assessing the plausibility and quality of generated HOI sequences. The Foot Skating Ratio (**FSR**) measures the proportion of frames exhibiting unrealistic foot sliding. The Penetration Ratio (**Pene**) calculates the average fraction of object vertices intersecting the human mesh across the sequence. The Contact Ratio (**Contact**) assesses the frequency with which the human and object maintain consistent contact throughout the motion. More details can be found in Sec. B.2 of the Appendix. Following Li et al. (2023b), we also report contact precision ($C_{prec}$), recall ($C_{rec}$), and F1 score ($C_{F1}$) to measure frame-wise contact accuracy, though these may not fully capture plausibility due to the diversity of text-to-HOI generation (see Sec. B.2 of the Appendix).

Existing methods (Peng et al., 2023; Diller & Dai, 2024; Wu et al., 2024a; Song et al., 2024) typically rely on feature extractors trained using relatively small-scale HOI datasets or their evaluator only measure human motion quality, which can negatively impact the robustness and reliability of evaluations. To address this limitation, we retrain our feature extraction models using the larger-scale InterAct dataset, leveraging regressed joint representations in conjunction with object BPS (Prokudin et al., 2019) features. Additional details can be found in Sec. B.1 of the Appendix.

Table 2: **Ablation study** of token-separation strategies on the InterAct dataset (Xu et al., 2025a). We report R-Precision with batch size 256.

| hand–body separation | human–object separation | R-Precision↑ | | | FID↓ | MM Dist↓ | Diversity→ | FSR↓ | Pene↓ | Contact→ | Interaction↑ | | |
|---|---|---|---|---|---|---|---|---|---|---|---|---|---|
| | | Top 1 | Top 2 | Top 3 | | | | | | | $C_{prec}$ | $C_{rec}$ | $C_{F1}$ |
| ✓ | ✓ | **0.421**±0.014 | **0.637**±0.016 | **0.754**±0.016 | **0.148**±0.014 | **2.756**±0.018 | 7.712±0.050 | 0.078±0.197 | 0.132±0.001 | 0.132±0.001 | 0.731±0.002 | **0.615**±0.016 | **0.627**±0.011 |
| – | ✓ | 0.414±0.001 | 0.611±0.003 | 0.717±0.007 | 0.157±0.017 | 2.863±0.009 | 7.749±0.125 | 0.089±0.186 | 0.131±0.000 | 0.124±0.003 | **0.750**±0.001 | 0.576±0.014 | 0.611±0.009 |
| ✓ | – | 0.409±0.003 | 0.590±0.005 | 0.693±0.002 | 0.155±0.010 | 2.917±0.030 | 7.768±0.067 | **0.063**±0.155 | **0.105**±0.002 | 0.107±0.004 | 0.742±0.004 | 0.524±0.009 | 0.572±0.007 |

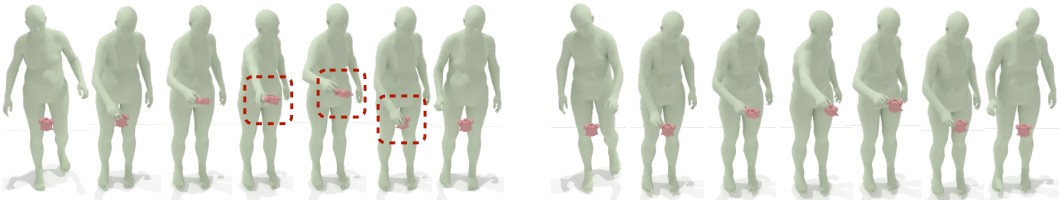

The person who stands near the alarm clock walks to it and uses the right hand to pick it up and see the time, and then puts it back and stretch the arms.

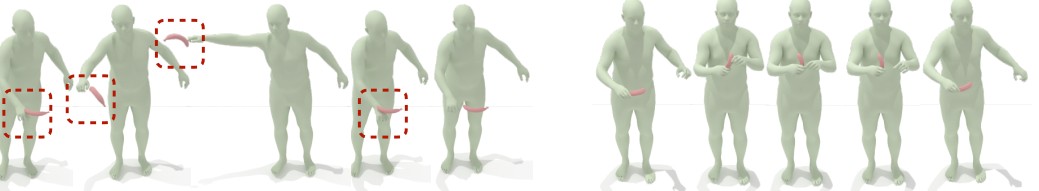

The individual stands next to a banana, approaches it, grasps it with the right hand and passes it forward before placing it down, then returns and stretches the arms.

Figure 4: **Qualitative comparison** between our method using body and hand merged into a single token (**left**) versus separating body and hand into distinct tokens (**right**). Unrealistic grasping artifacts produced by the single-token approach are highlighted in red dashed boxes. Our separate-token strategy yields better results.

**Implementation Details.** In the main paper, we fix the modality pairs by setting $m_1 = \{b, h\}$, $m_2 = \{o\}$. Additional experiments on all of the component combinations are presented in the Appendix C. Unless otherwise specified, we set the guidance weights as $\omega_1 = 0.5$, $\omega_2 = 3.0$ with a noise-level offset $\delta = 250$ while the total denoising steps $K = 500$. During training, object geometry is represented using a BPS (Prokudin et al., 2019) comprising 1024 points. Our models are trained on NVIDIA A100 GPUs, leveraging mixed-precision training with FP16 precision and flash attention. Training converges within approximately 24 hours using a single GPU. For the transformer backbone, we adopt an 8-layer transformer decoder structure with a latent dimension of 512 and a feed-forward dimension of 1024. Additional implementation details are in Sec. A.

**Baselines.** We compare against four recent HOI generation baselines. HOI-Diff (Peng et al., 2023), a diffusion-based framework for text-driven HOI, employs a transformer backbone (Tevet et al., 2022b) with classifier guidance via an affordance predictor (Dhariwal & Nichol, 2021). CHOIS (Li et al., 2023a), originally requiring inputs beyond text, is adapted here for text-only prompts. InterDiff (Xu et al., 2023b) is modified by replacing its historical motion encoder with a text encoder. Text2HOI (Cha et al., 2024) is incorporated for full-body HOI generation, following their protocol to train a static contact-map estimator. To assess our guidance strategy, we further evaluate two variants: (**i**) our method without guidance and (**ii**) our full model with LIGHT.

**Quantitative Evaluation.** As shown in Table 1, LIGHT surpasses prior diffusion-based text-to-HOI methods on most key metrics: it attains higher R-precision for tighter text-animation alignment and lower FID and MM Dist scores for better generation quality. Tables D and E demonstrate that our method surpass other baselines, when trained on small-scale datasets, mirroring baseline's setting. Sec. C demonstrate that our framework is able to generalize on two new tasks without retraining, with guidance showing improvement over counterpart without guidance, even on these unseen tasks.

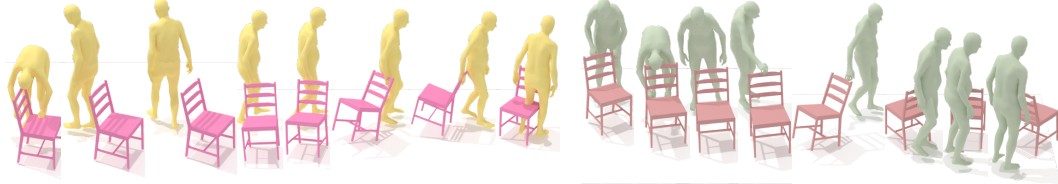

Push the woodchair, release the hands, then drag the woodchair, and set it back down.

Figure 5: **Qualitative comparison. Left**: our LIGHT *without* guidance. **Right**: our full method *with* guidance, which markedly enhances generation quality.

Table 3: **Ablation study.** We compare models trained *with* and *without* data augmentation on the InterAct dataset (Xu et al., 2025a). Experiments on unseen objects include in-category and cross-category objects never observed during training. We report R-Precision with batch size 256.

| Augmented | Unseen | R-Precision↑ | | | FID↓ | MM Dist↓ | Diversity→ | FSR↓ | Pene↓ | Contact→ | Interaction↑ | | |
|---|---|---|---|---|---|---|---|---|---|---|---|---|---|
| | | Top 1 | Top 2 | Top 3 | | | | | | | $C_{prec}$ | $C_{rec}$ | $C_{F1}$ |
| ✗ | In-category | $0.216^{\pm 0.004}$ | $0.316^{\pm 0.005}$ | $0.396^{\pm 0.001}$ | $0.151^{\pm 0.015}$ | $2.151^{\pm 0.033}$ | $6.347^{\pm 0.022}$ | $\mathbf{0.031}^{\pm 0.095}$ | $0.281^{\pm 0.013}$ | $0.193^{\pm 0.001}$ | $0.640^{\pm 0.008}$ | $0.503^{\pm 0.011}$ | $0.809^{\pm 0.006}$ |
| ✓ | In-category | $\mathbf{0.279}^{\pm 0.012}$ | $\mathbf{0.386}^{\pm 0.004}$ | $\mathbf{0.466}^{\pm 0.007}$ | $\mathbf{0.133}^{\pm 0.025}$ | $2.271^{\pm 0.006}$ | $\mathbf{6.696}^{\pm 0.045}$ | $0.059^{\pm 0.137}$ | $0.305^{\pm 0.016}$ | $0.190^{\pm 0.002}$ | $\mathbf{0.651}^{\pm 0.018}$ | $\mathbf{0.512}^{\pm 0.012}$ | $\mathbf{0.833}^{\pm 0.006}$ |
| ✗ | Cross-category | $0.022^{\pm 0.001}$ | $\mathbf{0.044}^{\pm 0.015}$ | $0.064^{\pm 0.016}$ | $5.118^{\pm 0.018}$ | $5.078^{\pm 0.060}$ | $\mathbf{5.655}^{\pm 0.055}$ | $0.226^{\pm 0.259}$ | $\mathbf{0.098}^{\pm 0.005}$ | $0.049^{\pm 0.001}$ | $0.388^{\pm 0.003}$ | $0.077^{\pm 0.002}$ | $0.351^{\pm 0.005}$ |
| ✓ | Cross-category | $0.022^{\pm 0.007}$ | $0.043^{\pm 0.003}$ | $\mathbf{0.071}^{\pm 0.009}$ | $\mathbf{4.924}^{\pm 0.046}$ | $\mathbf{4.788}^{\pm 0.032}$ | $5.509^{\pm 0.010}$ | $\mathbf{0.032}^{\pm 0.092}$ | $0.203^{\pm 0.025}$ | $0.116^{\pm 0.000}$ | $\mathbf{0.447}^{\pm 0.007}$ | $\mathbf{0.184}^{\pm 0.001}$ | $\mathbf{0.560}^{\pm 0.008}$ |

**Qualitative Evaluation.** In the baseline comparison (Figure 3), our method attains higher contact accuracy, whereas the baseline exhibits noticeable penetration and floating artifacts, and their finger articulation is often incorrect. We also conduct a user study to complement the qualitative validation, which is discussed in Sec. D, demonstrating that ours produces animation with higher quality.

**Impact of Pace-Induced Guidance.** We evaluate the impact of our proposed guidance. Figure 5 shows examples comparing HOI sequences generated with and without pace-induced guidance. With guidance enabled, generated animations clearly exhibit more precise contact dynamics and fewer unrealistic motion artifacts. This highlights that our approach effectively enhances the plausibility and overall visual quality of generated results. Table 1 corroborates these observations: combining guidance with our pipeline achieves the strongest performance.

**Impact of Separating Tokens.** In Figure 4, we show that separate hand token modeling yields noticeably more precise grasping motions and object placements. By comparison, the unified-token baseline often produces unnatural grasps and misaligned objects, underscoring the benefits of our separation strategy. Quantitative results in Table 2 reinforce this trend: token separation for three modalities with the our framework achieves superior performance across all metrics. We also conduct ablation on subset evaluation (Tables G and F), to mirror the setting of existing baselines (Li et al., 2023a; Xu et al., 2023b; Peng et al., 2023) trained on smaller dataset.

**Impact of Augmented Data.** Table A and Figure A illustrate that our augmentation framework generates realistic human-object interaction animations well suited for training. We evaluate models trained with augmented data under three distinct train-test splits: (i) Partitioned by sequence, consistent with the main experimental setup. Here, the test set contains no novel objects – all objects also appear in training. (ii) Partitioned by object category, where the test set consists of in-category objects unseen during training. (iii) Partitioned by object category, where the test set consists of cross-category objects entirely unseen during training. In all cases, the model is trained on a combination of the original training data and its in-category augmented data, then evaluated on the corresponding test set using our pretrained evaluator without fine-tuning. Table 3 shows that augmentation consistently improves performance across metrics on the unseen-object evaluation. This demonstrates that object-level augmentation substantially enhances generalizability, benefiting both in-category and cross-category unseen objects.

**Impact of Guidance Intensity and Denoising lagging.** Here, we investigate how the key hyperparameters, specifically $\omega_2$ and $\delta$, affect the effectiveness of our pace-induced guidance. As shown in Figure 6, varying the guidance weight $\omega_2$ and the denoising offset $\delta$ reveals a clear optimum. Weak guidance (small $\omega_2$) degrades FID; $\omega_2 = 4$ injects just enough prior for the best quality; larger $\omega_2$ over-constrains, reducing diversity and motion fluidity. Likewise, small offsets ($\delta < 100$) keep the staged and uniform branches too similar for effective correction, $\delta = 200$ strikes the best balance, and larger offsets might cause excessive divergence and abrupt fusion.

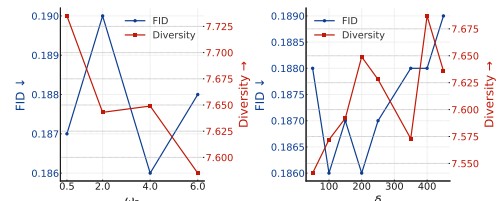

Figure 6: Ablation study on the schedule-based guidance weight $\omega_2$ and denoising preceding offset $\delta$. Left: $\delta$ fixed at 200 while varying $\omega_2$. Right: $\omega_2$ fixed at 4 while varying $\delta$.

**Analysis of Guidance Direction.** Motivated by the need to understand why guidance works, we follow (Rajabi et al., 2025) and analyze the directions induced by LIGHT and traditional CFG. Specifically, we measure how each guidance direction correlates with two reference directions:

Table 4: Comparison of the mean gradient similarity with penetration-descending direction, direction towards GT distribution. $\delta = 250$ is used and mean gradient similarity is calculated on all the guiding steps. we define $\langle \mathbf{a}, \mathbf{b} \rangle = \frac{\mathbf{a}}{\|\mathbf{a}\|} \cdot \frac{\mathbf{b}}{\|\mathbf{b}\|}$.

| Dataset | $<g_{CFG}, g_{GT}>$ | $<g_{LIGHT}, g_{GT}>$ | $<g_{CFG}, \nabla L_{pen}>$ | $<g_{LIGHT}, \nabla L_{pen}>$ |
|---|---|---|---|---|
| InterAct (Xu et al., 2025a) | 0.401 | 0.401 + 0.002 | 0.217 | 0.217 **+ 0.035** |
| OMOMO (Li et al., 2023a) | 0.395 | 0.395 + 0.029 | 0.239 | 0.239 **+ 0.032** |

(i) the displacement toward the ground-truth (GT) $g_{\text{GT}} = \hat{x}(0) - x_S(\lambda)$. where $\hat{x}(0)$ is defined in Sec. 3 as ground truth conterpart, and (ii) the penetration-reducing gradient $\nabla L_{\text{pen}}$. We provides comparisons between two produced guidance directions, where (i) the guidance direction of our LIGHT is defined as $g_{\text{LIGHT}} = \mathcal{G}_\theta(x'_S, \lambda', d) - \mathcal{G}_\theta(x_S(\lambda), \lambda, d)$, as defined in Equation 6. (ii) the text-cfg direction is $g_{\text{CFG}} = \mathcal{G}_\theta(x_U(\lambda), \lambda, d) - \mathcal{G}_\theta(x_U(\lambda), \lambda, \emptyset))$. as defined in Equation 2. We report the mean cosine similarity between two guidance directions and two reference directions. Empirically, our pace-induced guidance consistently steers motion toward the desired distribution, achieving GT-aligned similarity on par with text-conditioned CFG. To probe effects on low-level contact, we also measure the cosine similarity between two directions with and the penetration-reducing gradient $\nabla L_{\text{pen}}$. As shown in Table 4, our LIGHT shows more correlation with this gradient compared to pure text CFG. We hypothesize that the hard-dropout form of CFG biases samples toward the marginal data distribution – enhancing global plausibility but diminishing sensitivity to contact-specific cues. In LIGHT, the "unconditional" path retains weak conditioning via noisier, lagged components (*e.g.,* slight human-object surface misalignment). Then, the contrast between clean and noise branch will focus on surface-alignment signals and explicitly promotes depenetration and alignment.

## 5 CONCLUSION

We introduced LIGHT, a framework for human-object interaction animation that jointly models dynamic human motion and diverse object geometries without relying on hand-crafted priors or kinematic constraints. By decoupling modality-specific components with individualized noise schedules, the model enables cleaner elements to guide noisier ones, yielding inherently contact-aware generation. Generalization is further enhanced through synthetic object augmentation, which serves as a data prior to promote invariance of contact semantics across geometric variations. Collectively, these contributions establish a new form of guidance for HOI animation, providing both a scalable path toward more generalizable modeling and a natural extension of guidance within the diffusion forcing paradigm.

## ACKNOWLEDGMENTS

This work was supported in part by Snap Inc., NSF under Grants 2106825 and 2519216, the DARPA Young Faculty Award, the ONR Grant N00014-26-1-2099, and the NIFA Award 2020-67021-32799. This work used computational resources, including the NCSA Delta and DeltaAI and the PTI Jetstream2 supercomputers through allocations CIS230012, CIS230013, CIS240311, and CIS240428 from the Advanced Cyberinfrastructure Coordination Ecosystem: Services & Support (ACCESS) program, as well as the TACC Frontera supercomputer, Amazon Web Services (AWS), and OpenAI API through the National Artificial Intelligence Research Resource (NAIRR) Pilot.

## ETHICS STATEMENT

The realism and flexibility of animations produced by our approach could facilitate misuse, such as generating convincing yet fictitious scenarios involving human interactions with objects, which might lead to misinformation. To proactively mitigate these risks, we explicitly adopt abstract human-body representations, specifically, the SMPL model, which inherently lacks identifiable facial or biometric features. Thus, our approach substantially decreases the likelihood of synthesized content being exploited for identity misrepresentation or other privacy-invasive applications, thereby ensuring responsible and ethical usage.

## REPRODUCIBILITY STATEMENT

We provide a detailed description of the model architecture in Sec. 3. Details of adapted baselines are introduced in Sec. 4. For augmentation, we include all the details in Sec. A.2. Comprehensive implementation details, including hyperparameters, evaluation metrics, and the training procedure of the evaluator, are reported in Sec. 3 and further expanded in Sec. A, to facilitate reproducibility.

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

APPENDIX

In the Appendix, we provide additional methodological details and experimental results: (i) an expanded explanation of our loss formulation and object-augmentation procedure in Sec. A; (iii) definitions of the proposed metrics in Sec. B.2; and (iv) extra experimental results, omitted from the main paper for space, in Sec. D.

**LLM Usage.** We employ large language models (LLMs), such as ChatGPT, to assist in polishing our paper. Specifically, LLMs are used to correct grammatical errors, refine word choice, and improve overall fluency. We do not use LLM in formulating our methodology or running experiments

# A  ADDITIONAL DETAILS OF METHODOLOGY

## A.1  TRAINING LOSS

In training, we regularize our LIGHT with three loss terms, foot-skating loss ($L_{\text{fs}}$), velocity loss ($L_v$), and contact loss ($L_{\text{cont}}$). The total regularization objective is

$$L_{\text{reg}} = \lambda_{\text{fs}}\, L_{\text{fs}} + \lambda_{\text{v}}\, L_{\text{v}} + \lambda_{\text{cont}}\, L_{\text{cont}}, \tag{8}$$

where we set $\lambda_{\text{fs}} = 1, \lambda_{\text{v}} = 0.02, \lambda_{\text{cont}} = 0.1$.

**Foot-skating loss.** To constrain foot sliding during ground contact, we penalize the deviation between predicted and ground-truth foot velocities:

$$L_{\text{fs}} = \sum_{t=1}^{T} \sum_{f=1}^{4} c_t^f \left\| \left(\boldsymbol{j}_{t,f}^p - \boldsymbol{j}_{t-1,f}^p\right) - \left(\hat{\boldsymbol{j}}_{t,f}^p - \hat{\boldsymbol{j}}_{t-1,f}^p\right) \right\|_2^2, \tag{9}$$

where $c_t^f \in \{0, 1\}$ is the ground-truth foot-contact label for joint $f$ at time $t$, and $\hat{\boldsymbol{j}}_{t,f}^p$ (*resp.* $\hat{\boldsymbol{j}}_{t,f}^p$) denotes the predicted (*resp.* ground-truth) 3-D position of that joint. We follow (Guo et al., 2022a) for the foot-joint definitions and contact annotation.

**Velocity loss.** To encourage smooth temporal evolution of both human and object trajectories, we match first-order differences between predictions and ground truth:

$$
\begin{aligned}
L_{\text{v}} = {} & \lambda_{\text{pv}} \sum_{t=1}^{T} \sum_{j=1}^{J} \left\| \left(\boldsymbol{j}_{t,j}^p - \boldsymbol{j}_{t-1,j}^p\right) - \left(\hat{\boldsymbol{j}}_{t,j}^p - \hat{\boldsymbol{j}}_{t-1,j}^p\right) \right\|_2^2 \\
& + \lambda_{\text{otv}} \sum_{t=1}^{T} \left\| \left(\boldsymbol{o}_t^t - \boldsymbol{o}_{t-1}^t\right) - \left(\hat{\boldsymbol{o}}_t^t - \hat{\boldsymbol{o}}_{t-1}^t\right) \right\|_2^2 \\
& + \lambda_{\text{orv}} \sum_{t=1}^{T} \left\| \left(\boldsymbol{o}_t^r - \boldsymbol{o}_{t-1}^r\right) - \left(\hat{\boldsymbol{o}}_t^r - \hat{\boldsymbol{o}}_{t-1}^r\right) \right\|_2^2,
\end{aligned} \tag{10}
$$

where $\boldsymbol{j}_{t,j}^p$ is the position of human joint $j$, while $\boldsymbol{o}_t^t$ and $\boldsymbol{o}_t^r$ are the object translation and rotation at time $t$. We set $\lambda_{\text{pv}} = 1, \lambda_{\text{otv}} = 1, \lambda_{\text{orv}} = 1$.

**Contact loss.** To promote accurate human–object interactions, we minimize joint-to-surface distances whenever contact is expected:

$$L_{\text{cont}} = \sum_{t=1}^{T} \sum_{j=1}^{J} \left(d\left(\boldsymbol{j}_{t,j}^p,\, \hat{\boldsymbol{V}}_t^o\right) \hat{c}_t^j\right)^2, \tag{11}$$

where $d(\cdot, \cdot)$ returns the minimum Euclidean distance between joint $\hat{\boldsymbol{j}}_{t,j}^p$ and the set of ground truth object vertices $\boldsymbol{V}_t^o$, and $c_t^j$ is the ground-truth binary contact label for joint $j$ at time $t$, where we extract this information from ground truth data whether any joint to mesh distance is less than 0.03 m.

Table A: Quantitative evaluation of augmented data quality relative to ground truth annotations.

| Dataset | Data source | Pene$^\downarrow$ | Floating$^\downarrow$ | $CH_{prec}^\uparrow$ | $CH_{rec}^\uparrow$ | $CH_{F1}^\uparrow$ |
|---|---|---|---|---|---|---|
| InterAct (Xu et al., 2025a) | Ground Truth | 0.008 | 0.005 | 0.947 | 0.911 | 0.920 |
| | Augmentation | 0.010 | 0.011 | 0.980 | 0.980 | 0.980 |
| GRAB (Taheri et al., 2020) | Ground Truth | 0.003 | 0.000 | 1.000 | 1.000 | 1.000 |
| | Augmentation | 0.004 | 0.000 | 0.994 | 0.998 | 0.996 |

## A.2 ADDITIONAL DETAILS ON OBJECT AUGMENTATION

We employ an optimization-based strategy to enhance our dataset by transferring each original human-object interaction (HOI) sequence onto a novel object instance of the same category. Specifically, given an original HOI sequence with human motion and an object trajectory, we first train a dense correspondence network following Xie et al. (2024); Zhou et al. (2022b). Specifically, first, we extract the AABB (Axis-Aligned Bounding Box) of the object, and normalize to make the diagram of AABB to be 1, and the center of AABB to be at the origin. Then we train the corresponding network following Zhou et al. (2022b). The model architecture consists of a PointNet (Qi et al., 2017) and multilayer perceptron (MLP), which maps an unordered point cloud to the orientation. We follow Zhou et al. (2022b) to supervise the model with a chamfer distance loss. This network learns to map points on the source object's surface to corresponding points on a new object instance selected from a 3D shape repository in the same category, *e.g.*, ShapeNet (Chang et al., 2015) or Objaverse (Deitke et al., 2023). Using the learned correspondence, we replace the original object in the HOI sequence with the novel object, obtaining an initial transformed trajectory for the new object.

The goal is to position and move the new object such that the human's contact interactions remain consistent with the original sequence. In other words, for each contact constraint observed in the original sequence, *i.e.,* a specific human joint maintaining contact at frame with the original object, the corresponding surface point on the new object, identified via the correspondence network, should likewise remain in contact with that same human joint. We achieve this by optimizing the new object's pose parameters and the original human parameters over time to preserve the spatial alignment of these contact points while maintaining the physical plausibility of the interaction. To refine the new object's trajectory, we formulate an objective function that is a weighted sum of multiple alignment terms:

$$\mathcal{L} = \lambda_{\text{con}} L_{\text{con}} + \lambda_{\text{normal}} L_{\text{normal}} + \lambda_{\text{colli}} L_{\text{colli}} + \lambda_{\text{init}} L_{\text{init}} + \lambda_{\text{acc}} L_{\text{acc}}, \tag{12}$$

where $\lambda_{\text{con}}$, $\lambda_{\text{normal}}$, $\lambda_{\text{colli}}$, $\lambda_{\text{init}}$, and $\lambda_{\text{acc}}$ are weighting coefficients. Each loss term $L_*$ is designed to enforce a particular aspect of alignment between the human and the new object:

- $L_{\text{con}}$: penalizes any deviation in the distance between the human's contact points and the new object's corresponding surface points.
- $L_{\text{normal}}$: encourages the surface normals of the new object at contact regions to align with those of the source object, ensuring that the orientation of the object relative to the human's contacting limb (*e.g.*, a hand grasp) remains consistent with the original interaction.
- $L_{\text{colli}}$: penalizes interpenetration or unintended collisions between the human body and the new object, enforcing that the object remains outside the human geometry except at the intended contact regions.
- $L_{\text{init}}$: regularizes the solution towards the original motion.
- $L_{\text{acc}}$: promotes smooth motion by penalizing abrupt changes in velocity and acceleration.

By minimizing $\mathcal{L}$, we refine the human and novel object's trajectory so that the human's interaction with the object remains natural and consistent with the original sequence. Thus, the new object is placed and moved in such a way that its correspondence-defined contact points remain matched to the human's grasp or touch points throughout the sequence.

## B ADDITIONAL DETAILS OF EXPERIMENTAL SETUP

### B.1 EVALUATOR REPRESENTATION AND EVALUATOR TRAINING

Our evaluator takes as input the global human joint locations, object rotation, object translation, and object dynamic BPS. The dynamic BPS is computed using a static basis point set, following Li et al. (2023b); Xu et al. (2025a). Consequently, the evaluator assesses the quality of the entire HOI sequence

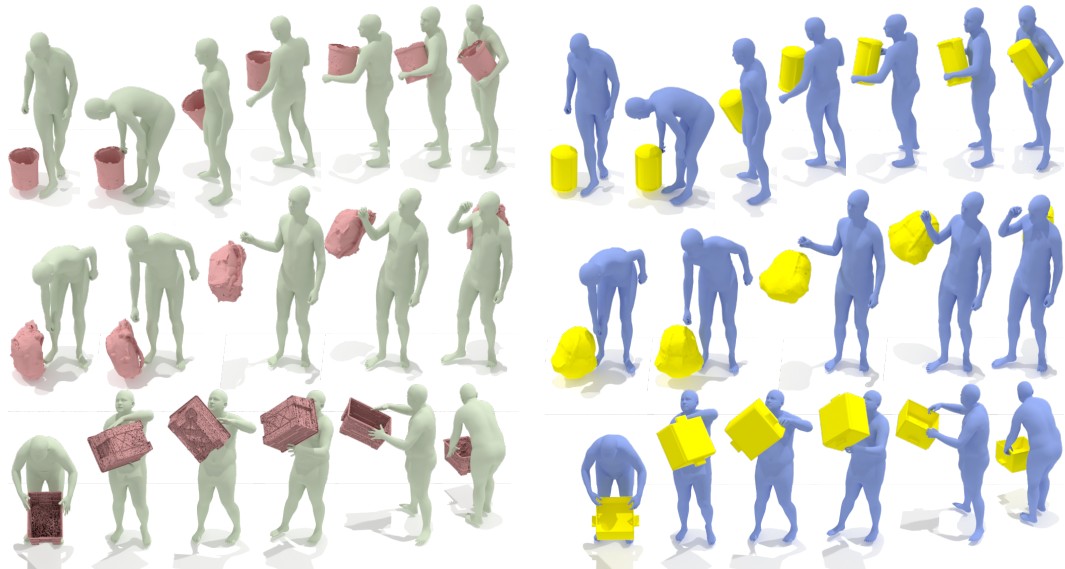

Figure A: We visualize our augmentation results, where original MoCap sequences (**left**) are enriched by introducing novel objects (**right**) sourced from ShapeNet (Chang et al., 2015) and Objaverse (Deitke et al., 2023). Through optimization, human and object motions are aligned at corresponding contact points.

rather than evaluating human or object motion in isolation. Following Lu et al. (2023); Petrovich et al. (2023), we jointly train the text and HOI motion encoders of our evaluator. Departing from traditional training on classification tasks (Guo et al., 2022a), we utilize a sequence-level contrastive learning objective based on the InfoNCE loss (Oord et al., 2018), following recent implementations (Lu et al., 2023; Petrovich et al., 2023). Our text encoding module incorporates Sentence-BERT (Reimers & Gurevych, 2019).

## B.2 Evaluation Metrics

We outline our metric definitions below. Standard text-to-motion metrics are not restated; we simply extend them with the object and contact representations introduced in the main paper. For their original formulations, see Guo et al. (2022a).

**Contact Percentage.** This metric measures how consistently the human's body joints stay in contact with the object, indicating the quality of physical interaction. Extending Li et al. (2023b) from hand contact to whole-body contact, we define

$$\text{Contact} \;=\; \frac{1}{T\,J} \sum_{t=1}^{T} \sum_{j=1}^{J} \mathbf{1}\Big( d\big(\boldsymbol{j}_{t,j}^{p}, \boldsymbol{V}_{t}^{o}\big) < \gamma \Big), \tag{13}$$

where $\boldsymbol{j}_{t,j}^{p} \in \mathbb{R}^3$ is the 3-D position of the $j$-th joint at time $t$, $\boldsymbol{V}_{t}^{o}$ is the set of object vertices at time $t$, and $d(\boldsymbol{j}_{t,j}^{p}, \boldsymbol{V}_{t}^{o})$ is the minimum distance from the joint to the object surface. We use the contact threshold $\gamma = 0.05$ following Li et al. (2023a). The more alignment of values of Equation equation 13 with the ground truth indicates better quality.

**Frame-Wise Contact Matching.** Following Li et al. (2023b), we report frame-wise contact matching, where $C_{F1}$, $C_{prec}$, and $C_{rec}$ denote the F1 score, precision, and recall of detected contacts between hands and objects. We use a contact threshold of $\gamma = 0.05$, consistent with Li et al. (2023a). However, this metric is not entirely suitable for the text-to-HOI task, since frame-level alignment is not strictly enforced. In generative tasks conditioned only on text, it is standard practice to evaluate outputs with distributional metrics (*e.g.,* FID in text-to-motion or text-to-image), rather than element- or frame-wise measures such as MPJPE or MSE. Nevertheless, we include frame-wise contact scores as they provide a partial indication of contact quality.

Table B: **Quantitative evaluation** of unconditional generation on the BEHAVE dataset (Bhatnagar et al., 2022). For R-Precision we adopt a batch size of 64. We don't retrain our model for unconditional generation.

| $m_1$ | $m_2$ | FID$\downarrow$ | Diversity$\rightarrow$ | FSR$\downarrow$ | Pene$\downarrow$ | Contact$\rightarrow$ | Interaction$\uparrow$ | | |
|---|---|---|---|---|---|---|---|---|---|
| | | | | | | | $C_{prec}$ | $C_{rec}$ | $C_{F1}$ |
| $-$ | $-$ | $0.989^{\pm 0.231}$ | $6.678^{\pm 0.163}$ | $\mathbf{0.165}^{\pm 0.214}$ | $0.123^{\pm 0.004}$ | $0.162^{\pm 0.001}$ | $0.640^{\pm 0.007}$ | $0.610^{\pm 0.013}$ | $0.585^{\pm 0.003}$ |
| $o, h$ | $b$ | $0.958^{\pm 0.229}$ | $6.715^{\pm 0.141}$ | $0.172^{\pm 0.221}$ | $0.122^{\pm 0.011}$ | $0.164^{\pm 0.003}$ | $0.643^{\pm 0.005}$ | $0.612^{\pm 0.024}$ | $0.585^{\pm 0.008}$ |
| $b, h$ | $o$ | $0.957^{\pm 0.258}$ | $6.672^{\pm 0.224}$ | $\mathbf{0.165}^{\pm 0.217}$ | $0.122^{\pm 0.005}$ | $0.167^{\pm 0.001}$ | $0.640^{\pm 0.003}$ | $0.617^{\pm 0.013}$ | $0.588^{\pm 0.003}$ |
| $b, o$ | $h$ | $0.963^{\pm 0.240}$ | $6.694^{\pm 0.180}$ | $0.173^{\pm 0.221}$ | $0.122^{\pm 0.006}$ | $0.169^{\pm 0.002}$ | $\mathbf{0.646}^{\pm 0.013}$ | $0.622^{\pm 0.020}$ | $\mathbf{0.593}^{\pm 0.002}$ |
| $b$ | $o, h$ | $0.976^{\pm 0.238}$ | $6.691^{\pm 0.201}$ | $0.172^{\pm 0.222}$ | $0.122^{\pm 0.009}$ | $0.170^{\pm 0.004}$ | $0.640^{\pm 0.009}$ | $\mathbf{0.624}^{\pm 0.023}$ | $\mathbf{0.593}^{\pm 0.005}$ |
| $o$ | $b, h$ | $\mathbf{0.932}^{\pm 0.157}$ | $\mathbf{6.757}^{\pm 0.175}$ | $0.175^{\pm 0.221}$ | $0.124^{\pm 0.002}$ | $0.176^{\pm 0.001}$ | $0.630^{\pm 0.001}$ | $0.622^{\pm 0.026}$ | $0.586^{\pm 0.005}$ |
| $h$ | $b, o$ | $0.968^{\pm 0.218}$ | $6.708^{\pm 0.184}$ | $0.172^{\pm 0.221}$ | $\mathbf{0.121}^{\pm 0.004}$ | $0.166^{\pm 0.003}$ | $0.641^{\pm 0.007}$ | $0.616^{\pm 0.023}$ | $0.586^{\pm 0.005}$ |

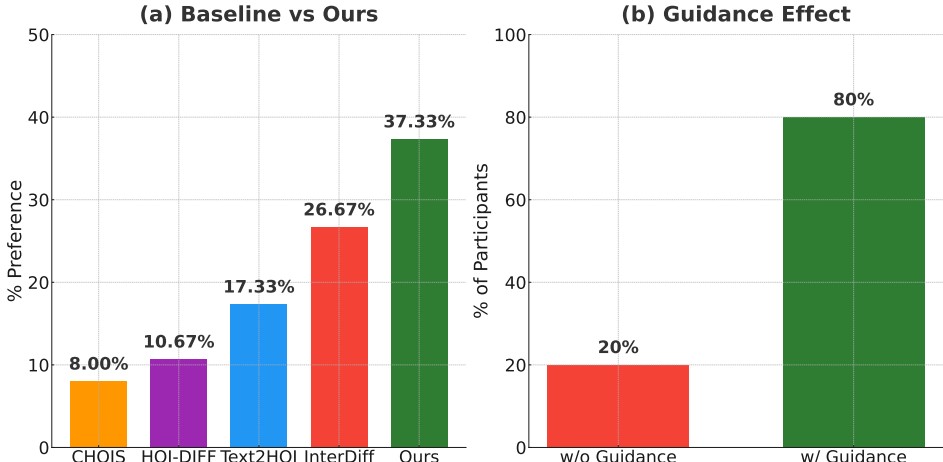

Figure B: **User study results.** 15 participants evaluates our method comparing baselines, and our guidance mechanism. Percentages indicate participant preference.

**Penetration.** Penetration quantifies geometric violations between the human mesh and the object. After reconstructing the full human mesh via forward kinematics, we compute

$$\text{Penetration} \; = \; \frac{1}{T} \sum_{t=1}^{T} \sum_{i=1}^{N} \left| \min\!\big(\text{sdf}^{h_t}(\boldsymbol{v}_{o_i}), \, 0\big) \right|, \tag{14}$$

where $\text{sdf}^{h_t}(\boldsymbol{v}_{o_i})$ is the signed distance from object vertex $\boldsymbol{v}_{o_i}$ to the human mesh at time $t$. Negative SDF values indicate penetration depth; clamping positive values to zero ensures that only inter-penetrations are accumulated. Lower scores in Eq. equation 14 correspond to fewer or shallower penetrations.

**Foot-Skating Ratio (FSR).** FSR gauges how well the character's feet remain stationary during ground-contact phases, a key indicator of motion realism. Following Karunratanakul et al. (2023), we compute

$$\text{FSR} \; = \; \frac{1}{T} \sum_{t=1}^{T} c_t^F \, \mathbf{1}\!\Big( \big\| (\boldsymbol{j}_{t,f}^p - \boldsymbol{j}_{t-1,f}^p) \cdot \text{fps} \big\| < \epsilon \Big), \tag{15}$$

where $c_t^F \in \{0, 1\}$ is the ground-contact indicator for a foot joint at time $t$, $\boldsymbol{j}_{t,f}^p$ is that foot's position, and fps converts frame-to-frame displacement to velocity. We adopt $\epsilon = 0.025$ and a ground-contact height threshold of 0.05 as in Karunratanakul et al. (2023). FSR ranges from 0 to 1; higher values signify less foot sliding and more realistic foot-ground interaction.

## C  ADDITIONAL ANALYSIS OF GUIDANCE

**Quantitative Evaluation of Our Guidance Across Tasks.** In addition to the Text-to-HOI task presented in the main paper, we evaluate our guidance under different task settings. Specifically, we consider (i) *unconditional HOI generation*, where the model takes as input an object mesh and a

Table C: **Ablation** of modality combinations $m_1, m_2$ on InterAct (Xu et al., 2025a), BEHAVE (Bhatnagar et al., 2022), and OMOMO (Li et al., 2023a) datasets. We report R-Precision with batch size 256.

| Dataset | $\omega_1$ | $m_1$ | $m_2$ | R-Precision$^\uparrow$ Top 1 | Top 2 | Top 3 | FID$^\downarrow$ | MM Dist$^\downarrow$ | Diversity$^\rightarrow$ | FSR$^\downarrow$ | Pene$^\downarrow$ | Contact$^\rightarrow$ | Interaction$^\uparrow$ $C_{prec}$ | $C_{rec}$ | $C_{F1}$ |
|---|---|---|---|---|---|---|---|---|---|---|---|---|---|---|---|
| BEHAVE | 0.5 | – | – | $0.258^{\pm0.000}$ | $0.447^{\pm0.003}$ | $0.543^{\pm0.027}$ | $0.493^{\pm0.028}$ | $3.297^{\pm0.080}$ | $6.878^{\pm0.139}$ | $\mathbf{0.164}^{\pm0.239}$ | $0.144^{\pm0.004}$ | $0.162^{\pm0.002}$ | $\mathbf{0.737}^{\pm0.017}$ | $0.711^{\pm0.015}$ | $0.694^{\pm0.015}$ |
| | | $o,h$ | $b$ | $0.275^{\pm0.003}$ | $0.455^{\pm0.003}$ | $0.535^{\pm0.022}$ | $0.485^{\pm0.028}$ | $3.315^{\pm0.070}$ | $6.942^{\pm0.133}$ | $0.170^{\pm0.244}$ | $0.150^{\pm0.003}$ | $0.164^{\pm0.001}$ | $0.732^{\pm0.010}$ | $0.712^{\pm0.014}$ | $0.693^{\pm0.013}$ |
| | | $b,h$ | $o$ | $0.254^{\pm0.005}$ | $0.451^{\pm0.003}$ | $0.537^{\pm0.030}$ | $0.491^{\pm0.017}$ | $3.305^{\pm0.065}$ | $6.892^{\pm0.133}$ | $0.165^{\pm0.238}$ | $0.146^{\pm0.002}$ | $0.163^{\pm0.002}$ | $0.732^{\pm0.012}$ | $0.710^{\pm0.013}$ | $0.691^{\pm0.013}$ |
| | | $b,o$ | $h$ | $0.268^{\pm0.003}$ | $0.449^{\pm0.016}$ | $0.543^{\pm0.022}$ | $0.480^{\pm0.032}$ | $3.308^{\pm0.065}$ | $6.941^{\pm0.127}$ | $0.170^{\pm0.245}$ | $0.144^{\pm0.002}$ | $0.167^{\pm0.001}$ | $0.728^{\pm0.011}$ | $0.713^{\pm0.018}$ | $0.694^{\pm0.016}$ |
| | | $b$ | $o,h$ | $0.268^{\pm0.003}$ | $0.453^{\pm0.005}$ | $0.545^{\pm0.019}$ | $\mathbf{0.478}^{\pm0.087}$ | $3.314^{\pm0.069}$ | $6.951^{\pm0.125}$ | $0.169^{\pm0.245}$ | $0.148^{\pm0.002}$ | $0.167^{\pm0.002}$ | $0.727^{\pm0.010}$ | $0.714^{\pm0.019}$ | $0.694^{\pm0.016}$ |
| | | $o$ | $b,h$ | $\mathbf{0.277}^{\pm0.011}$ | $\mathbf{0.475}^{\pm0.014}$ | $\mathbf{0.561}^{\pm0.041}$ | $0.481^{\pm0.087}$ | $\mathbf{3.277}^{\pm0.076}$ | $\mathbf{6.961}^{\pm0.066}$ | $0.171^{\pm0.245}$ | $\mathbf{0.140}^{\pm0.004}$ | $0.170^{\pm0.001}$ | $0.731^{\pm0.006}$ | $\mathbf{0.720}^{\pm0.011}$ | $\mathbf{0.697}^{\pm0.007}$ |
| | | $h$ | $b,o$ | $0.266^{\pm0.016}$ | $0.451^{\pm0.003}$ | $0.527^{\pm0.027}$ | $0.483^{\pm0.024}$ | $3.326^{\pm0.063}$ | $6.948^{\pm0.130}$ | $0.169^{\pm0.244}$ | $0.149^{\pm0.001}$ | $0.165^{\pm0.001}$ | $0.730^{\pm0.007}$ | $0.712^{\pm0.017}$ | $0.693^{\pm0.014}$ |
| (Bhatnagar et al., 2022) | 0.0 | – | – | $0.225^{\pm0.008}$ | $0.430^{\pm0.011}$ | $\mathbf{0.518}^{\pm0.003}$ | $0.500^{\pm0.008}$ | $\mathbf{3.379}^{\pm0.033}$ | $6.920^{\pm0.220}$ | $\mathbf{0.163}^{\pm0.232}$ | $0.143^{\pm0.009}$ | $0.158^{\pm0.001}$ | $0.722^{\pm0.013}$ | $0.693^{\pm0.013}$ | $0.676^{\pm0.010}$ |
| | | $o,h$ | $b$ | $0.227^{\pm0.011}$ | $0.422^{\pm0.016}$ | $0.508^{\pm0.011}$ | $0.495^{\pm0.016}$ | $3.382^{\pm0.033}$ | $6.938^{\pm0.217}$ | $0.168^{\pm0.237}$ | $0.142^{\pm0.008}$ | $0.161^{\pm0.000}$ | $0.723^{\pm0.014}$ | $0.699^{\pm0.018}$ | $0.681^{\pm0.015}$ |
| | | $b,h$ | $o$ | $0.223^{\pm0.005}$ | $0.430^{\pm0.011}$ | $\mathbf{0.518}^{\pm0.003}$ | $0.493^{\pm0.009}$ | $3.391^{\pm0.043}$ | $6.945^{\pm0.199}$ | $0.166^{\pm0.234}$ | $0.140^{\pm0.001}$ | $0.162^{\pm0.001}$ | $\mathbf{0.728}^{\pm0.002}$ | $\mathbf{0.701}^{\pm0.001}$ | $\mathbf{0.682}^{\pm0.000}$ |
| | | $b,o$ | $h$ | $\mathbf{0.232}^{\pm0.019}$ | $0.422^{\pm0.032}$ | $0.504^{\pm0.005}$ | $\mathbf{0.467}^{\pm0.021}$ | $3.394^{\pm0.015}$ | $6.959^{\pm0.193}$ | $0.169^{\pm0.236}$ | $0.140^{\pm0.004}$ | $0.163^{\pm0.002}$ | $0.721^{\pm0.016}$ | $0.700^{\pm0.018}$ | $0.679^{\pm0.013}$ |
| | | $b$ | $o,h$ | $0.230^{\pm0.005}$ | $0.416^{\pm0.003}$ | $0.508^{\pm0.000}$ | $0.479^{\pm0.021}$ | $3.394^{\pm0.032}$ | $\mathbf{6.971}^{\pm0.187}$ | $0.168^{\pm0.237}$ | $0.139^{\pm0.007}$ | $0.163^{\pm0.000}$ | $0.723^{\pm0.013}$ | $\mathbf{0.701}^{\pm0.013}$ | $0.680^{\pm0.012}$ |
| | | $o$ | $b,h$ | $0.229^{\pm0.014}$ | $0.414^{\pm0.000}$ | $0.512^{\pm0.000}$ | $0.491^{\pm0.012}$ | $3.384^{\pm0.033}$ | $6.967^{\pm0.205}$ | $0.171^{\pm0.239}$ | $0.139^{\pm0.009}$ | $0.164^{\pm0.002}$ | $0.726^{\pm0.014}$ | $0.699^{\pm0.017}$ | $0.680^{\pm0.013}$ |
| | | $h$ | $b,o$ | $0.219^{\pm0.000}$ | $0.416^{\pm0.024}$ | $0.500^{\pm0.011}$ | $0.471^{\pm0.025}$ | $3.408^{\pm0.022}$ | $6.968^{\pm0.179}$ | $0.170^{\pm0.237}$ | $\mathbf{0.139}^{\pm0.006}$ | $0.161^{\pm0.001}$ | $0.720^{\pm0.015}$ | $0.698^{\pm0.017}$ | $0.678^{\pm0.013}$ |
| InterAct | 0.5 | – | – | $0.426^{\pm0.001}$ | $0.642^{\pm0.014}$ | $0.751^{\pm0.010}$ | $0.160^{\pm0.007}$ | $\mathbf{2.722}^{\pm0.005}$ | $7.722^{\pm0.053}$ | $\mathbf{0.067}^{\pm0.173}$ | $0.131^{\pm0.004}$ | $0.121^{\pm0.008}$ | $0.738^{\pm0.004}$ | $0.574^{\pm0.022}$ | $0.601^{\pm0.019}$ |
| | | $o,h$ | $b$ | $0.423^{\pm0.000}$ | $0.636^{\pm0.014}$ | $0.749^{\pm0.013}$ | $0.154^{\pm0.015}$ | $2.741^{\pm0.005}$ | $7.723^{\pm0.057}$ | $0.073^{\pm0.189}$ | $0.131^{\pm0.001}$ | $0.120^{\pm0.007}$ | $0.741^{\pm0.005}$ | $0.567^{\pm0.018}$ | $0.594^{\pm0.020}$ |
| | | $b,h$ | $o$ | $0.423^{\pm0.003}$ | $0.635^{\pm0.014}$ | $0.751^{\pm0.018}$ | $0.153^{\pm0.009}$ | $2.736^{\pm0.058}$ | $7.732^{\pm0.058}$ | $0.070^{\pm0.180}$ | $\mathbf{0.129}^{\pm0.000}$ | $0.121^{\pm0.007}$ | $0.741^{\pm0.001}$ | $0.574^{\pm0.014}$ | $0.601^{\pm0.017}$ |
| | | $b,o$ | $h$ | $\mathbf{0.433}^{\pm0.010}$ | $\mathbf{0.644}^{\pm0.016}$ | $\mathbf{0.757}^{\pm0.011}$ | $0.154^{\pm0.016}$ | $2.751^{\pm0.001}$ | $7.721^{\pm0.033}$ | $0.078^{\pm0.197}$ | $\mathbf{0.129}^{\pm0.008}$ | $0.120^{\pm0.002}$ | $\mathbf{0.742}^{\pm0.012}$ | $0.563^{\pm0.011}$ | $0.593^{\pm0.011}$ |
| | | $b$ | $o,h$ | $0.423^{\pm0.007}$ | $0.635^{\pm0.011}$ | $0.751^{\pm0.018}$ | $0.156^{\pm0.011}$ | $2.736^{\pm0.010}$ | $7.723^{\pm0.056}$ | $0.074^{\pm0.191}$ | $0.131^{\pm0.001}$ | $0.121^{\pm0.005}$ | $0.738^{\pm0.002}$ | $0.566^{\pm0.016}$ | $0.592^{\pm0.015}$ |
| | | $o$ | $b,h$ | $0.421^{\pm0.014}$ | $0.637^{\pm0.016}$ | $0.754^{\pm0.016}$ | $\mathbf{0.148}^{\pm0.014}$ | $2.756^{\pm0.018}$ | $7.712^{\pm0.050}$ | $0.078^{\pm0.197}$ | $0.132^{\pm0.001}$ | $0.132^{\pm0.001}$ | $0.731^{\pm0.002}$ | $\mathbf{0.615}^{\pm0.014}$ | $\mathbf{0.627}^{\pm0.011}$ |
| | | $h$ | $b,o$ | $0.426^{\pm0.005}$ | $0.635^{\pm0.008}$ | $0.746^{\pm0.007}$ | $0.156^{\pm0.012}$ | $2.751^{\pm0.000}$ | $\mathbf{7.709}^{\pm0.058}$ | $0.074^{\pm0.189}$ | $\mathbf{0.125}^{\pm0.002}$ | $0.121^{\pm0.006}$ | $0.741^{\pm0.001}$ | $0.568^{\pm0.007}$ | $0.595^{\pm0.009}$ |
| (Xu et al., 2025a) | 0.0 | – | – | $\mathbf{0.395}^{\pm0.012}$ | $\mathbf{0.599}^{\pm0.009}$ | $\mathbf{0.715}^{\pm0.004}$ | $0.196^{\pm0.014}$ | $2.885^{\pm0.004}$ | $7.708^{\pm0.041}$ | $\mathbf{0.066}^{\pm0.165}$ | $0.121^{\pm0.003}$ | $0.123^{\pm0.010}$ | $0.739^{\pm0.008}$ | $0.567^{\pm0.019}$ | $0.599^{\pm0.018}$ |
| | | $o,h$ | $b$ | $0.387^{\pm0.005}$ | $0.589^{\pm0.001}$ | $0.701^{\pm0.005}$ | $\mathbf{0.184}^{\pm0.018}$ | $2.947^{\pm0.001}$ | $7.684^{\pm0.049}$ | $0.074^{\pm0.181}$ | $0.125^{\pm0.003}$ | $0.121^{\pm0.012}$ | $0.737^{\pm0.013}$ | $0.552^{\pm0.024}$ | $0.586^{\pm0.022}$ |
| | | $b,h$ | $o$ | $0.389^{\pm0.007}$ | $0.589^{\pm0.002}$ | $0.701^{\pm0.001}$ | $0.188^{\pm0.014}$ | $2.921^{\pm0.006}$ | $7.702^{\pm0.057}$ | $0.067^{\pm0.168}$ | $0.121^{\pm0.005}$ | $0.124^{\pm0.011}$ | $0.742^{\pm0.013}$ | $0.568^{\pm0.028}$ | $0.601^{\pm0.027}$ |
| | | $b,o$ | $h$ | $0.387^{\pm0.009}$ | $0.589^{\pm0.012}$ | $0.703^{\pm0.003}$ | $0.190^{\pm0.025}$ | $2.938^{\pm0.002}$ | $7.700^{\pm0.042}$ | $0.071^{\pm0.177}$ | $\mathbf{0.120}^{\pm0.006}$ | $0.121^{\pm0.010}$ | $0.737^{\pm0.003}$ | $0.553^{\pm0.021}$ | $0.587^{\pm0.019}$ |
| | | $b$ | $o,h$ | $0.389^{\pm0.008}$ | $0.588^{\pm0.005}$ | $0.696^{\pm0.001}$ | $0.185^{\pm0.025}$ | $2.957^{\pm0.008}$ | $\mathbf{7.672}^{\pm0.056}$ | $0.076^{\pm0.185}$ | $0.122^{\pm0.004}$ | $0.119^{\pm0.011}$ | $0.741^{\pm0.010}$ | $0.548^{\pm0.033}$ | $0.584^{\pm0.026}$ |
| | | $o$ | $b,h$ | $0.389^{\pm0.013}$ | $0.590^{\pm0.012}$ | $0.699^{\pm0.003}$ | $0.186^{\pm0.007}$ | $2.942^{\pm0.002}$ | $7.707^{\pm0.067}$ | $0.074^{\pm0.183}$ | $0.134^{\pm0.006}$ | $0.134^{\pm0.005}$ | $0.741^{\pm0.008}$ | $\mathbf{0.614}^{\pm0.004}$ | $\mathbf{0.628}^{\pm0.001}$ |
| | | $h$ | $b,o$ | $0.388^{\pm0.006}$ | $0.589^{\pm0.007}$ | $0.696^{\pm0.005}$ | $0.184^{\pm0.019}$ | $2.946^{\pm0.002}$ | $7.683^{\pm0.050}$ | $0.073^{\pm0.179}$ | $0.121^{\pm0.002}$ | $0.122^{\pm0.011}$ | $\mathbf{0.745}^{\pm0.017}$ | $0.559^{\pm0.021}$ | $0.593^{\pm0.021}$ |
| OMOMO | 0.5 | – | – | $0.301^{\pm0.003}$ | $0.538^{\pm0.009}$ | $0.702^{\pm0.015}$ | $0.103^{\pm0.017}$ | $1.663^{\pm0.023}$ | $\mathbf{7.432}^{\pm0.050}$ | $\mathbf{0.012}^{\pm0.042}$ | $0.101^{\pm0.004}$ | $0.193^{\pm0.000}$ | $\mathbf{0.942}^{\pm0.001}$ | $0.852^{\pm0.007}$ | $0.882^{\pm0.005}$ |
| | | $o,h$ | $b$ | $0.299^{\pm0.003}$ | $0.538^{\pm0.012}$ | $0.702^{\pm0.007}$ | $0.102^{\pm0.020}$ | $1.672^{\pm0.007}$ | $7.441^{\pm0.047}$ | $0.013^{\pm0.045}$ | $0.099^{\pm0.004}$ | $0.193^{\pm0.000}$ | $0.942^{\pm0.000}$ | $0.854^{\pm0.006}$ | $0.884^{\pm0.004}$ |
| | | $b,h$ | $o$ | $0.301^{\pm0.003}$ | $\mathbf{0.540}^{\pm0.012}$ | $0.706^{\pm0.009}$ | $0.103^{\pm0.020}$ | $1.669^{\pm0.001}$ | $7.439^{\pm0.042}$ | $0.013^{\pm0.046}$ | $\mathbf{0.098}^{\pm0.001}$ | $0.195^{\pm0.000}$ | $0.941^{\pm0.000}$ | $0.854^{\pm0.005}$ | $0.883^{\pm0.004}$ |
| | | $b,o$ | $h$ | $0.298^{\pm0.007}$ | $\mathbf{0.540}^{\pm0.009}$ | $0.705^{\pm0.008}$ | $0.099^{\pm0.017}$ | $1.668^{\pm0.010}$ | $\mathbf{7.432}^{\pm0.054}$ | $0.013^{\pm0.045}$ | $0.099^{\pm0.003}$ | $0.194^{\pm0.000}$ | $0.940^{\pm0.000}$ | $0.852^{\pm0.009}$ | $0.882^{\pm0.005}$ |
| | | $b$ | $o,h$ | $\mathbf{0.302}^{\pm0.001}$ | $0.538^{\pm0.015}$ | $\mathbf{0.707}^{\pm0.011}$ | $0.099^{\pm0.018}$ | $\mathbf{1.657}^{\pm0.010}$ | $7.443^{\pm0.043}$ | $0.013^{\pm0.045}$ | $0.100^{\pm0.001}$ | $0.194^{\pm0.000}$ | $0.940^{\pm0.000}$ | $0.851^{\pm0.008}$ | $0.881^{\pm0.005}$ |
| | | $o$ | $b,h$ | $0.296^{\pm0.001}$ | $0.532^{\pm0.007}$ | $0.699^{\pm0.014}$ | $0.100^{\pm0.022}$ | $1.677^{\pm0.023}$ | $7.451^{\pm0.044}$ | $0.013^{\pm0.045}$ | $0.099^{\pm0.001}$ | $0.196^{\pm0.003}$ | $0.936^{\pm0.001}$ | $\mathbf{0.856}^{\pm0.009}$ | $\mathbf{0.883}^{\pm0.005}$ |
| | | $h$ | $b,o$ | $0.300^{\pm0.004}$ | $0.539^{\pm0.014}$ | $0.705^{\pm0.008}$ | $0.100^{\pm0.022}$ | $1.667^{\pm0.001}$ | $7.445^{\pm0.042}$ | $0.013^{\pm0.046}$ | $0.099^{\pm0.001}$ | $0.193^{\pm0.000}$ | $0.941^{\pm0.000}$ | $0.855^{\pm0.007}$ | $0.884^{\pm0.005}$ |
| (Li et al., 2023a) | 0.0 | – | – | $0.279^{\pm0.004}$ | $\mathbf{0.514}^{\pm0.011}$ | $\mathbf{0.673}^{\pm0.018}$ | $0.132^{\pm0.003}$ | $1.811^{\pm0.009}$ | $7.395^{\pm0.059}$ | $\mathbf{0.012}^{\pm0.043}$ | $0.104^{\pm0.007}$ | $0.192^{\pm0.000}$ | $\mathbf{0.940}^{\pm0.000}$ | $0.844^{\pm0.003}$ | $0.875^{\pm0.002}$ |
| | | $o,h$ | $b$ | $0.273^{\pm0.003}$ | $0.510^{\pm0.003}$ | $0.670^{\pm0.014}$ | $0.131^{\pm0.032}$ | $\mathbf{1.821}^{\pm0.008}$ | $7.393^{\pm0.063}$ | $\mathbf{0.012}^{\pm0.044}$ | $0.104^{\pm0.009}$ | $0.191^{\pm0.000}$ | $\mathbf{0.940}^{\pm0.000}$ | $0.844^{\pm0.004}$ | $0.875^{\pm0.002}$ |
| | | $b,h$ | $o$ | $\mathbf{0.281}^{\pm0.008}$ | $\mathbf{0.514}^{\pm0.003}$ | $0.668^{\pm0.005}$ | $\mathbf{0.129}^{\pm0.036}$ | $1.815^{\pm0.006}$ | $\mathbf{7.400}^{\pm0.047}$ | $\mathbf{0.012}^{\pm0.043}$ | $\mathbf{0.102}^{\pm0.006}$ | $0.191^{\pm0.000}$ | $0.938^{\pm0.002}$ | $0.847^{\pm0.003}$ | $0.877^{\pm0.003}$ |
| | | $b,o$ | $h$ | $0.277^{\pm0.003}$ | $0.511^{\pm0.007}$ | $0.668^{\pm0.014}$ | $\mathbf{0.129}^{\pm0.032}$ | $1.823^{\pm0.011}$ | $7.389^{\pm0.065}$ | $\mathbf{0.012}^{\pm0.043}$ | $0.104^{\pm0.011}$ | $0.191^{\pm0.000}$ | $0.939^{\pm0.002}$ | $0.842^{\pm0.004}$ | $0.874^{\pm0.003}$ |
| | | $b$ | $o,h$ | $0.274^{\pm0.004}$ | $\mathbf{0.514}^{\pm0.003}$ | $0.667^{\pm0.012}$ | $0.126^{\pm0.029}$ | $1.833^{\pm0.008}$ | $7.395^{\pm0.041}$ | $\mathbf{0.012}^{\pm0.043}$ | $0.101^{\pm0.008}$ | $0.190^{\pm0.001}$ | $0.937^{\pm0.003}$ | $0.841^{\pm0.001}$ | $0.873^{\pm0.001}$ |
| | | $o$ | $b,h$ | $0.278^{\pm0.004}$ | $0.507^{\pm0.004}$ | $0.665^{\pm0.012}$ | $0.125^{\pm0.039}$ | $1.841^{\pm0.012}$ | $7.411^{\pm0.067}$ | $\mathbf{0.012}^{\pm0.042}$ | $0.101^{\pm0.012}$ | $0.192^{\pm0.005}$ | $0.939^{\pm0.006}$ | $\mathbf{0.852}^{\pm0.012}$ | $\mathbf{0.882}^{\pm0.006}$ |
| | | $h$ | $b,o$ | $0.274^{\pm0.009}$ | $\mathbf{0.514}^{\pm0.000}$ | $0.669^{\pm0.015}$ | $\mathbf{0.129}^{\pm0.032}$ | $1.834^{\pm0.000}$ | $7.403^{\pm0.056}$ | $\mathbf{0.012}^{\pm0.044}$ | $0.102^{\pm0.009}$ | $0.191^{\pm0.000}$ | $0.938^{\pm0.002}$ | $0.844^{\pm0.001}$ | $0.876^{\pm0.001}$ |

human shape to synthesize a complete HOI sequence without extra conditioning, and (ii) *controllable HOI generation*, where the model is conditioned on a richer set of inputs—including the full object motion sequence, object mesh, human shape, and a textual description—to generate the corresponding HOI sequence. For both settings, we reuse the model from the main paper without retraining. This is made possible by our independent noise scheduling, which allows the model to noised out unused conditioning signals or overlap one denoising modality with input condition. Table B demonstrates that our guidance remains effective across multiple tasks.

**Analysis of Different Component Combinations.** In this section, we experiment with multiple component combinations. Table C demonstrates that our guidance is effective on flexible component combinations.

# D    OTHER ADDITIONAL EXPERIMENTAL RESULTS

**User Study.** We conducted a user study with 15 participants. Among them, 86.7% reported that our augmented data exhibits fewer artifacts than the ground truth (GT), while 100% agreed that the interaction consistency between GT and augmented sequences is preserved. Additional statistics are provided in Figure B, which further demonstrate that (**i**) our method qualitatively outperforms the baseline, and (**ii**) the proposed guidance mechanism leads to improved performance.

**Quantitative Evaluation of the Quality of Augmented Data.** Table A demonstrates that our augmented data can match the quality of the ground-truth set, showing minimal contact errors and demonstrating its suitability as synthetic training data. And our augmentation of small objects on GRAB (Taheri et al., 2020) could also greatly align the contact of the ground truth dataset

**Quantitative Evaluation on additional datasets.** The Table D, E and ablations in Table F, G, demonstrates that our guidance method is effective across multiple datasets, rather than only on a large-scale dataset. And also, token separation and independent noise schedule are also effective.

**Disscussion of Runtime efficiency.** Our guidance requires three model forward passes per step, and a complete inference procedure is needed before applying guidance. Specifically, for a batch of 64 samples, each sample consisting of 300 frames, our guided inference completes in approximately 72 seconds on one A100 gpu. In contrast, HOI-Diff, InterDiff both takes about 15 seconds without guidance.

Table D: **Quantitative comparisons** on the OMOMO dataset (Li et al., 2023a) between our method and baseline approaches. We report R-Precision with batch size 256.

| Method | R-Precision↑ | | | FID↓ | MM Dist↓ | Diversity→ | FSR↓ | Pene↓ | Contact→ | Interaction↑ | | |
|---|---|---|---|---|---|---|---|---|---|---|---|---|
| | Top 1 | Top 2 | Top 3 | | | | | | | $C_{prec}$ | $C_{rec}$ | $C_{F1}$ |
| Ground Truth | $0.318^{\pm0.003}$ | $0.560^{\pm0.001}$ | $0.731^{\pm0.001}$ | $0.000^{\pm0.000}$ | $1.499^{\pm0.000}$ | $7.400^{\pm0.047}$ | $0.012^{\pm0.039}$ | $0.067^{\pm0.000}$ | $0.262^{\pm0.000}$ | $1.000^{\pm0.000}$ | $1.000^{\pm0.000}$ | $1.000^{\pm0.000}$ |
| HOI-Diff Peng et al. (2023) | $0.221^{\pm0.005}$ | $0.403^{\pm0.001}$ | $0.542^{\pm0.001}$ | $0.987^{\pm0.023}$ | $2.520^{\pm0.022}$ | $7.248^{\pm0.066}$ | $0.021^{\pm0.058}$ | $0.131^{\pm0.005}$ | $0.135^{\pm0.002}$ | $0.862^{\pm0.000}$ | $0.696^{\pm0.000}$ | $0.739^{\pm0.001}$ |
| CHOIS Li et al. (2023a) | $0.291^{\pm0.000}$ | $0.515^{\pm0.009}$ | $0.676^{\pm0.008}$ | $0.132^{\pm0.016}$ | $1.746^{\pm0.002}$ | $7.417^{\pm0.104}$ | $0.019^{\pm0.053}$ | $\mathbf{0.096}^{\pm0.004}$ | $0.230^{\pm0.001}$ | $0.937^{\pm0.000}$ | $\mathbf{0.900}^{\pm0.004}$ | $\mathbf{0.910}^{\pm0.002}$ |
| InterDiff Xu et al. (2023b) | $\mathbf{0.312}^{\pm0.001}$ | $\mathbf{0.548}^{\pm0.004}$ | $\mathbf{0.718}^{\pm0.004}$ | $0.163^{\pm0.009}$ | $\mathbf{1.587}^{\pm0.006}$ | $7.387^{\pm0.045}$ | $0.025^{\pm0.062}$ | $0.100^{\pm0.002}$ | $0.206^{\pm0.000}$ | $\mathbf{0.940}^{\pm0.016}$ | $0.863^{\pm0.016}$ | $0.890^{\pm0.017}$ |
| Text2HOI Cha et al. (2024) | $0.286^{\pm0.001}$ | $0.513^{\pm0.001}$ | $0.655^{\pm0.001}$ | $0.168^{\pm0.000}$ | $1.832^{\pm0.006}$ | $7.359^{\pm0.034}$ | $0.020^{\pm0.056}$ | $0.127^{\pm0.002}$ | $0.163^{\pm0.000}$ | $0.919^{\pm0.001}$ | $0.757^{\pm0.002}$ | $0.807^{\pm0.002}$ |
| LIGHT (**Ours**) w/o guidance | $0.279^{\pm0.003}$ | $0.514^{\pm0.011}$ | $0.673^{\pm0.018}$ | $0.132^{\pm0.031}$ | $1.811^{\pm0.009}$ | $\mathbf{7.395}^{\pm0.059}$ | $\mathbf{0.012}^{\pm0.043}$ | $0.104^{\pm0.007}$ | $0.192^{\pm0.000}$ | $\mathbf{0.940}^{\pm0.000}$ | $0.844^{\pm0.003}$ | $0.875^{\pm0.002}$ |
| LIGHT (**Ours**) w/ guidance | $0.302^{\pm0.004}$ | $0.538^{\pm0.015}$ | $0.707^{\pm0.011}$ | $\mathbf{0.099}^{\pm0.018}$ | $1.657^{\pm0.010}$ | $7.443^{\pm0.043}$ | $0.013^{\pm0.045}$ | $0.100^{\pm0.001}$ | $0.194^{\pm0.000}$ | $\mathbf{0.940}^{\pm0.000}$ | $0.851^{\pm0.008}$ | $0.881^{\pm0.005}$ |

Table E: **Quantitative comparisons** on the BEHAVE dataset Bhatnagar et al. (2022) between our method and baseline approaches. We report R-Precision with batch size 256.

| Method | R-Precision↑ | | | FID↓ | MM Dist↓ | Diversity→ | FSR↓ | Pene↓ | Contact→ | Interaction↑ | | |
|---|---|---|---|---|---|---|---|---|---|---|---|---|
| | Top 1 | Top 2 | Top 3 | | | | | | | $C_{prec}$ | $C_{rec}$ | $C_{F1}$ |
| Ground Truth | $0.559^{\pm0.005}$ | $0.910^{\pm0.000}$ | $0.926^{\pm0.000}$ | $-0.000^{\pm0.000}$ | $1.483^{\pm0.018}$ | $6.940^{\pm0.153}$ | $0.181^{\pm0.224}$ | $0.094^{\pm0.001}$ | $0.204^{\pm0.001}$ | $1.000^{\pm0.000}$ | $1.000^{\pm0.000}$ | $1.000^{\pm0.000}$ |
| HOI-Diff Peng et al. (2023) | $\mathbf{0.413}^{\pm0.010}$ | $\mathbf{0.624}^{\pm0.009}$ | $\mathbf{0.740}^{\pm0.011}$ | $0.689^{\pm0.031}$ | $\mathbf{3.029}^{\pm0.007}$ | $7.620^{\pm0.085}$ | $\mathbf{0.072}^{\pm0.160}$ | $\mathbf{0.103}^{\pm0.010}$ | $0.084^{\pm0.006}$ | $0.722^{\pm0.002}$ | $0.447^{\pm0.026}$ | $0.501^{\pm0.022}$ |
| CHOIS Li et al. (2023a) | $0.219^{\pm0.000}$ | $0.396^{\pm0.003}$ | $0.492^{\pm0.022}$ | $0.705^{\pm0.027}$ | $3.423^{\pm0.045}$ | $6.863^{\pm0.061}$ | $0.191^{\pm0.253}$ | $0.128^{\pm0.011}$ | $0.170^{\pm0.003}$ | $0.717^{\pm0.018}$ | $0.662^{\pm0.000}$ | $0.650^{\pm0.005}$ |
| InterDiff Xu et al. (2023b) | $0.258^{\pm0.027}$ | $0.443^{\pm0.019}$ | $0.531^{\pm0.027}$ | $0.630^{\pm0.033}$ | $3.219^{\pm0.040}$ | $6.862^{\pm0.066}$ | $0.208^{\pm0.264}$ | $0.143^{\pm0.011}$ | $0.173^{\pm0.005}$ | $0.723^{\pm0.001}$ | $0.682^{\pm0.013}$ | $0.676^{\pm0.004}$ |
| Text2HOI Cha et al. (2024) | $0.135^{\pm0.003}$ | $0.242^{\pm0.000}$ | $0.322^{\pm0.008}$ | $1.291^{\pm0.073}$ | $3.815^{\pm0.026}$ | $6.544^{\pm0.013}$ | $0.177^{\pm0.233}$ | $0.179^{\pm0.003}$ | $0.128^{\pm0.001}$ | $0.703^{\pm0.011}$ | $0.579^{\pm0.007}$ | $0.587^{\pm0.008}$ |
| LIGHT (**Ours**) w/o guidance | $0.225^{\pm0.008}$ | $0.430^{\pm0.011}$ | $0.518^{\pm0.003}$ | $0.500^{\pm0.015}$ | $3.379^{\pm0.033}$ | $\mathbf{6.920}^{\pm0.220}$ | $0.163^{\pm0.232}$ | $0.143^{\pm0.009}$ | $0.158^{\pm0.001}$ | $0.722^{\pm0.013}$ | $0.693^{\pm0.013}$ | $0.676^{\pm0.010}$ |
| LIGHT (**Ours**) w/ guidance | $0.277^{\pm0.011}$ | $0.475^{\pm0.014}$ | $0.561^{\pm0.041}$ | $\mathbf{0.481}^{\pm0.087}$ | $3.277^{\pm0.076}$ | $6.961^{\pm0.066}$ | $0.171^{\pm0.245}$ | $0.140^{\pm0.004}$ | $0.170^{\pm0.001}$ | $\mathbf{0.731}^{\pm0.006}$ | $\mathbf{0.720}^{\pm0.011}$ | $\mathbf{0.697}^{\pm0.007}$ |

Table F: **Ablation study** of token-separation strategies on the BEHAVE dataset Xu et al. (2025a). We report R-Precision with batch size 256.

| hand–body separation | human–object separation | R-Precision↑ | | | FID↓ | MM Dist↓ | Diversity→ | FSR↓ | Pene↓ | Contact→ | Interaction↑ | | |
|---|---|---|---|---|---|---|---|---|---|---|---|---|---|
| | | Top 1 | Top 2 | Top 3 | | | | | | | $C_{prec}$ | $C_{rec}$ | $C_{F1}$ |
| ✓ | ✓ | $0.277^{\pm0.011}$ | $0.475^{\pm0.014}$ | $0.561^{\pm0.041}$ | $\mathbf{0.481}^{\pm0.087}$ | $3.277^{\pm0.076}$ | $\mathbf{6.961}^{\pm0.066}$ | $0.171^{\pm0.245}$ | $0.140^{\pm0.004}$ | $0.170^{\pm0.001}$ | $\mathbf{0.731}^{\pm0.006}$ | $\mathbf{0.720}^{\pm0.011}$ | $\mathbf{0.697}^{\pm0.007}$ |
| − | ✓ | $\mathbf{0.297}^{\pm0.005}$ | $\mathbf{0.533}^{\pm0.008}$ | $\mathbf{0.643}^{\pm0.003}$ | $0.515^{\pm0.005}$ | $\mathbf{2.991}^{\pm0.053}$ | $6.963^{\pm0.068}$ | $0.173^{\pm0.250}$ | $\mathbf{0.126}^{\pm0.003}$ | $0.157^{\pm0.004}$ | $\mathbf{0.731}^{\pm0.001}$ | $0.701^{\pm0.012}$ | $0.692^{\pm0.006}$ |
| ✓ | − | $0.273^{\pm0.022}$ | $0.471^{\pm0.035}$ | $0.574^{\pm0.038}$ | $0.498^{\pm0.014}$ | $3.162^{\pm0.000}$ | $6.898^{\pm0.022}$ | $\mathbf{0.164}^{\pm0.246}$ | $0.135^{\pm0.000}$ | $0.166^{\pm0.001}$ | $0.713^{\pm0.006}$ | $0.690^{\pm0.006}$ | $0.676^{\pm0.005}$ |

Table G: **Ablation study** of token-separation strategies on the OMOMO dataset (Li et al., 2023a). We report R-Precision with batch size 256.

| hand–body separation | human–object separation | R-Precision↑ | | | FID↓ | MM Dist↓ | Diversity→ | FSR↓ | Pene↓ | Contact→ | Interaction↑ | | |
|---|---|---|---|---|---|---|---|---|---|---|---|---|---|
| | | Top 1 | Top 2 | Top 3 | | | | | | | $C_{prec}$ | $C_{rec}$ | $C_{F1}$ |
| ✓ | ✓ | $0.302^{\pm0.004}$ | $0.538^{\pm0.015}$ | $0.707^{\pm0.011}$ | $0.099^{\pm0.018}$ | $1.657^{\pm0.010}$ | $7.443^{\pm0.043}$ | $\mathbf{0.013}^{\pm0.045}$ | $0.100^{\pm0.001}$ | $0.194^{\pm0.000}$ | $\mathbf{0.940}^{\pm0.000}$ | $0.851^{\pm0.008}$ | $\mathbf{0.881}^{\pm0.005}$ |
| − | ✓ | $0.320^{\pm0.000}$ | $0.551^{\pm0.005}$ | $0.711^{\pm0.005}$ | $\mathbf{0.093}^{\pm0.008}$ | $\mathbf{1.483}^{\pm0.006}$ | $\mathbf{7.366}^{\pm0.071}$ | $0.017^{\pm0.004}$ | $\mathbf{0.062}^{\pm0.006}$ | $0.223^{\pm0.005}$ | $0.840^{\pm0.003}$ | $0.775^{\pm0.005}$ | $0.798^{\pm0.005}$ |
| ✓ | − | $\mathbf{0.322}^{\pm0.003}$ | $\mathbf{0.553}^{\pm0.003}$ | $\mathbf{0.715}^{\pm0.000}$ | $0.106^{\pm0.003}$ | $1.499^{\pm0.013}$ | $7.378^{\pm0.193}$ | $0.031^{\pm0.004}$ | $0.079^{\pm0.002}$ | $0.236^{\pm0.003}$ | $0.835^{\pm0.003}$ | $\mathbf{0.790}^{\pm0.000}$ | $\mathbf{0.804}^{\pm0.003}$ |

**Limitations.** Despite the encouraging results, our approach currently faces several limitations. First, our framework is designed specifically for single-object scenarios, and extending it to multi-object interactions remains an open challenge. Furthermore, the current model does not explicitly incorporate static environmental contexts or detailed scene geometry, potentially limiting the contextual realism of generated motions. Additionally, while our method improves interaction coherence and physical plausibility, it may still struggle with highly dynamic or physically complex interactions requiring precise and rapid manipulations. Finally, due to the computational complexity introduced by staged conditioning and diffusion forcing, real-time or interactive applications might be challenging.

