# OpenReview forum: "Unleashing Guidance Without Classifiers for Human-Object Interaction Animation"
_ICLR.cc/2026/Conference — ICLR 2026 Poster_

### Official Review · Reviewer_wUBQ · 2025-10-31

**Soundness:** 3
**Presentation:** 2
**Contribution:** 3
**Rating:** 2
**Confidence:** 4

**Summary:**

This paper proposes a new method for text-driven HOI generation. The main technical contributions are two fold. First, it porposes a new guidance mechanism where asynchronous denoising induces guidance without using an external classifier/objective function. Second, a contact-aware shape-spectrum data augmentation strategy is proposed that preserves contact semantics while varying object geometry. Experimental results on InterAct, BEHAVE, and OMOMO are reported.

**Strengths:**

1. The proposed data augmentation technique is promising. Generalization of the interaction semantics to similar objects is a fundamental challenge, which is not only useful for the HOI generation task studied in this paper, but may also be useful say other domains, say robotics.

2. State-of-the-art results are reported in this paper.

**Weaknesses:**

1. While the autors have explained **how** the proposed asynchronous denoising of different parts works for HOI generation, they didn't explain **why** it works. In a high-level sense, it works in a way analogous to CFG, but it lacks detailed analysis and investigation. Especially considering the proposed approach is inspired by (or connected to) the duffions forcing mechanism, which is a generic framework (not tailored for HOI generation), showing **insights** why it is useful for HOI generation is critical as future work may better understand when it works and when it may not.

2. The contact-aware shape-spectrum augmentation is a big contribution of this paper. But it lacks sufficient details of how it works (e.g., statistics of number of objects before and after augmentation) and no visual examples are shown in the main paper. Only limited illustrations are provided in the appendix and supplementary video. It makes readers hard to gauge the effectiveness of this part and reproduce it.

**Questions:**

1. $\textbf{x}_S$ is not defined in the paper. Could you please explain it?

2. In Eq. (4), for $\textbf{x}_S'$, it it a concatenation of $\textbf{x}_U^{m_1}$ and $\textbf{x}_S^{M_2}$? By the way, the symbol of $\textbf{x}$ in $\textbf{x}_S'$ is different from that in $\textbf{x}_U^{m_1}$ and $\textbf{x}_S^{M_2}$ in the paper.

3. Is ithe pace-induced guidance used in the inference only?

---

> ### Author Response · Authors · 2025-11-25
>
> We thank the reviewer for thoughtful and insightful comments. We address the concerns below:
>
>
> W1: Why the proposed asynchronous denoising works (HOI-specific insights)
> - The key reason our guidance is effective for HOI is that the task naturally involves two coupled modalities – human motion and object motion. Our asynchronous (pace-induced) guidance implicitly decouples these during denoising: the staged branch provides a cleaner reference, while the uniform branch carries a slightly lagged, noisier human-object configuration. The discrepancy between them allows one modality to regularize the other, and in practical we find it acting like a soft prior that encourages depenetration and better contact, even without a classifier related to such prior.
>
> Our paper already analyzed why this works, following CFG-style analyses [1, 2]:
> * **When the guidance works.** As in [1], Figure 5 (main) and Table D (Appendix) show that guidance is effective when the displacement between the staged and uniform branches (controlled by $\delta$ and $\omega_2$) lies in a suitable range – large enough to create a useful contrast, but not so large that the branches decorrelate. This regime yields clear quality gains.
> * **How it improves HOI-specific quality.** Following [2]'s analysis, Table 4 and the section *Analysis of Guidance Direction* show that our guidance direction brings larger improvements in depenetration than traditional text-conditioned CFG. This is illustrated in L467-L473 explaining that our guidance's gradient direction is better aligned with the penetration-reducing gradient.
> Based on this, we hypothesize that hard-dropout CFG biases samples toward the marginal data distribution, improving global plausibility. In contrast, LIGHT’s “unconditional” path retains noisier, human-object configurations (e.g., with slight surface misalignment). The contrast with the clean branch naturally amplifies surface-alignment signals, promoting depenetration and better local contact, as also illustrated in supplementary video 01:53-01:57.
>
> W2: Lack of details of how augmentation works, and lack of visual illustrations
> - We would like to clarify that we enrich the dataset from 217 objects to 1121 objects, as stated in L318-L319 in the original main paper.
> - We also provided a rich gallery of the augmented data in the original supplementary video 00:19-00:27, containing 15 examples of the augmented data. We further include 48 additional examples of the augmented data, and 8 additional comparisons between augmentation and ground truth in the revised supplementary materials (**augmentation.mp4**). Also, 2 additional generated examples from training with augmentation are included in the revised supplementary materials (**additional_examples.mp4**).
> - To demonstrate effectiveness, in addition to the visual results, we quantitatively evaluated the quality of the augmented data in Appendix Table A in the original submission, demonstrating that training with our augmented data improves the model’s generalization.
> - For reproduction, we will release the code and data upon acceptance. And we have also included all the implementation details in Appendix B.2.
> - We would like to politely ask whether you feel any specific details are missing, and we would be happy to clarify them further.
>
> Q1: Explain $\boldsymbol{x}_S$
> - Thanks for pointing this out. For the explanation of $\boldsymbol{x}_S$, we include it in the revision (please refer to Equation 7).
>
> Q2: Clarification of Eq. (4), and symbol difference
> - Thanks for pointing this out. For Equation 4, it is a concatenation. For the symbol difference, we have corrected it in the revision (please refer to Equation 4).
>
> Q3: Is the pace-induced guidance used in the inference only?
> - Yes.
>
>
> We thank the reviewer for the constructive feedback. If the reviewer has any further questions or suggestions, we are happy to discuss them
>
> [1] Rajabi, Javad, et al. Token Perturbation Guidance for Diffusion Models. NeurIPS, 2025.
>
> [2] Ho, Jonathan, et al. Classifier-free diffusion guidance. arXix:2207.12598, 2022.
>
> [3] Xie, Xianghui, et al. Intertrack: Tracking human object interaction without object templates. 3DV, 2025.

---

### Official Review · Reviewer_woqV · 2025-11-01

**Soundness:** 3
**Presentation:** 4
**Contribution:** 3
**Rating:** 6
**Confidence:** 4

**Summary:**

1. This paper presents LIGHT, an asynchronous-denoising–based guidance mechanism that achieves soft, flexible conditioning without relying on external classifiers.

2. In addition, the paper introduces contact-aware shape-spectrum augmentation, which maintains contact semantics while altering object geometry, enhancing robustness and generalization.

3. Extensive experiments further verify the approach, showing consistent improvements over existing baselines and enabling the generation of vivid, realistic interactions.

**Strengths:**

1. It proposes a pace-induced guidance mechanism to generate more realistic and plausible human-object interactions.

2. Through extensive experiments, it analyzes the effects of pace-induced guidance, token separation, augmented data, guidance intensity and denoising lag, and guidance direction.

3. The authors also provide fair comparisons by re-implementing and modifying prior baselines (e.g., InterDiff, Text2HOI).

**Weaknesses:**

1. There is no reference to Figure 5 in the section Impact of Guidance Intensity and Denoising Lagging. Please explicitly link the analysis to the figure.

2. In the Impact of Guidance Intensity and Denoising Lagging experiment, the paper states that the best value of δ is 300, but Figure 5 seems to suggest that 200 performs best. Could the authors clarify which value is correct?

3. I am not fully clear about the settings in Table 2. For the case where hand-body separation is ✓ and human-object separation is -, is the object token concatenated with the hand stream or with the body stream? In other words, which grouping is correct: {b, ho} or {bo, h}?

4. Additionally, if there are no separator tokens, meaning all tokens are combined, then it seems that the staged schedule cannot be applied. Could the authors clarify how the model behaves in this case?

**Questions:**

1. Will the authors release the code and the augmented data used for the Contact-Aware Shape-Spectrum Augmentation?

2. In Section 4, how do you compute x′_S from x^{m1}_U and x^{m2}_S? Is it a simple concatenation or another operation?

3. In the supplementary video at timestamp 1:06, the motions for InterDiff and Text2HOI appear very similar to each other but different from the proposed method. Were different text prompts used for these baseline results?
Please check it.

---

> ### Author Response · Authors · 2025-11-25
>
> We thank the reviewer for thoughtful and insightful comments. We address the concerns below:
>
> W1: No reference to Figure 5
> - Thanks for pointing this out. We have linked the analysis to Figure 5 in the revision (please refer to L444-L455).
>
>
> W2: Clarify which $\delta$  is the best
> - Thanks for pointing this out. We would like to clarify that 200 is the best value, as shown in Figure 5. We apologize for the confusion, and have corrected it in the revision (please refer to L444-L455).
>
> W3: Clarification on ablation setting
> - The object token is concatenated with the body stream, which is {bo, h}.
>
> W4: How does the model work without separator tokens?
> - Unlike the mask and separator tokens commonly used in masked transformers and autoregressive models, our diffusion denoiser employs bidirectional attention together with modality-wise positional encodings to indicate the positions of different modalities, following [1]. As a result, under the staged schedule, the attention mechanism remains aware of positional information.
>
> Q1: Code and data release
> - Yes. We will release data and code upon acceptance.
>
> Q2: Explain $\boldsymbol{x}'_S$
> - We use simple concatenation.
>
> Q3: Were different text prompts used in one visual example?
> - Thanks for the careful attention to the baseline comparisons. We carefully checked and the same text prompt was used. The possible reason why the motions generated by these two methods are similar in this example is that the baselines fail to generate motion aligned with the text, and instead produce a motion that appears more frequently in GRAB, which is passing an object.
>
> We thank the reviewer for the constructive feedback. If the reviewer has any further questions or suggestions, we are happy to discuss them
>
>
> [1] Cha, Junuk, et al. Text2hoi: Text-guided 3d motion generation for hand-object interaction. CVPR, 2024.

---

### Official Review · Reviewer_sZdf · 2025-11-01

**Soundness:** 2
**Presentation:** 2
**Contribution:** 2
**Rating:** 4
**Confidence:** 5

**Summary:**

The paper proposes a novel classifier-free guidance framework for diffusion-based HOI animation. Unlike prior methods relying on external contact classifiers or kinematic constraints, LIGHT achieves guidance through asynchronous denoising schedules. The cleaner (less noisy) modalities guide noisier ones, producing contact-aware behavior. Additionally, the paper introduces contact-aware shape-spectrum augmentation using ShapeNet and Objaverse objects to improve geometric generalization. Extensive experiments on the InterAct, BEHAVE, and OMOMO datasets demonstrate that LIGHT outperforms baselines such as HOI-Diff, InterDiff, CHOIS, and Text2HOI in FID, contact quality, and text-motion alignment, while maintaining realism and generalization across unseen object categories

**Strengths:**

- The idea of pace-induced guidance (asynchronous denoising between modalities) is innovative, extending diffusion forcing into a practical HOI setting
- The quantitative results are comprehensive, with thorough comparisons against existing baselines and well-conducted ablation studies.

**Weaknesses:**

- From the final visual results, the proposed strategy indeed demonstrates the ability to effectively leverage human priors to generate plausible interaction poses. However, the method still struggles with fine-grained contact modeling, and noticeable artifacts remain at the contact level.
 - The evaluation metrics seem somewhat questionable — the R-Precision scores are all within a similar range, and several other metrics also show minor differences across methods. It is unclear whether these metrics are sensitive enough to effectively distinguish the quality of human–object interactions.
- Since the video format is the most effective medium for evaluating the quality of human–object interactions, here the visual results are quite limited, with only two baseline comparisons, one augmented-training example, and one ablation study shown. This makes it hard to visually assess the claimed improvements or the model’s generalization capability.

**Questions:**

- Just curious about why does the prediction-based method Interdiff with modification (text-conditioning) achieve such competitive results?
- Some large-scale subsets of the InterAct dataset, such as BEHAVE and OMOMO, lack explicit hand motion (showing the mean hand pose). In this case, is the proposed hand–object separation strategy still meaningful or effective? Additionally, does the data augmentation stage include any hand-related synthesis to compensate for the missing hand motion?

---

> ### Author Response · Authors · 2025-11-25
>
> We thank the reviewer for thoughtful and insightful comments. We address the concerns below:
>
> W1: Noticeable artifacts remain at the contact level.
> - We would like to clarify that our method **does not rely on** any post-processing, classifier-based guidance for penetration reduction [1, 2, 4], or physics simulations [8] to correct artifacts. Compared with baselines that also do not use these tools, our approach, introducing guidance without classifiers, achieves noticeably better visual quality with fewer artifacts, with reduced penetration, improved contact, and less floating. It is generally expected that purely kinematics-based generation, without post-optimization or physics simulation, may exhibit minor artifacts.
> - Some of the observed artifacts stem partly from the dataset itself. Since the training data contains minor instances of contact floating and penetration, models inevitably learn these imperfections. As a result, similar artifacts are also present in existing works [1, 4].
> - If possible, could you kindly let us know which specific example you were referring to regarding the artifacts? We would be glad to offer further explanation on that.
>
> W2: Questionable metrics: R-precision within small ranges, metrics showing minor differences
> - We would like to clarify that for all evaluation metrics reported in this paper, we adopt **exactly** the same settings as in previous works [1, 2, 5].
> - We would clarify that the R-precision is effective, as the metric still captures meaningful **statistical differences**, and is not **saturated**.
>   - In Table A, we report both the mean and variance, which allows performance differences to be interpreted statistically rather than by absolute score values alone. The observed gaps between methods are also substantially larger than the variance, indicating that these differences are statistically meaningful.
>   - Our R-precision evaluation follows the benchmark protocol in [5], which uses a batch size of 64, and the values reported there are within a similar range to ours. We further test the effectiveness of our evaluator and metric by re-evaluating Tables 1, 2, C, D, E, F, G, and H in the paper with a larger batch size of 256, following [7], and these additional results are included in the corresponding Tables in the revision. As shown in the updated tables, the expanded numerical range confirms that R-precision is not saturated, and highlights even more substantial gains achieved by our method.
>   - Our evaluation of text-motion alignment is also strengthened by metrics like MM-DIST, which provides clear numerical distinctions across methods.
>
> W3: Limited video examples of comparisons
>
> - In the original supplementary video, we provided a rich gallery that demonstrates the generated motion by our method in 00:08-00:16 and a gallery that demonstrates our augmented data in 00:19-00:27.
> We have revised the supplementary materials to include more comparisons with the baselines, results of training with augmented data in **additional_examples.mp4**. Also we have included additional examples of augmented data in **augmentation.mp4**.
> We would appreciate your feedback on whether these additional examples sufficiently address your concern. If you have any preference regarding the visual examples, we would be happy to provide additional results during the discussion period.

---

> ### Author Response · Authors · 2025-11-25
>
> Q1: Why adapted InterDiff has strong performance?
> - A similar observation was also reported in previous work [3], where InterDiff was adapted as text-conditioned and achieved strong performance.
> - We hypothesize that InterDiff uses local attention instead of global attention used in vanilla Transformers, which restricts each token to attend only to a limited neighborhood or window around it. This design choice may improve motion smoothness and emphasize the relationship between neighbor frames. Also, with the past-frame motion encoder replaced by a text encoder, the adapted InterDiff architecture becomes similar to other MDM-based baselines and therefore could be reasonably expected to reach similar performance.
>
> Q2: Is hand-object separation effective on BEHAVE and OMOMO? Is hand-related synthesis included in augmentation?
> - We would like to clarify that InterAct [5] already corrects the hand motion for the OMOMO [2] and BEHAVE [8] subsets, with other additional subsets containing ground truth hand poses.
> - We would like to refer the reviewer to Tables G and H, which show that the hand-object separation strategy remains effective in these subset evaluations. This suggests that the approach is generally applicable for modeling whole-body motion, as treating body and hand motion as separate tokens allows the model to capture their distinct kinematic patterns and interaction dynamics more accurately.
> - The data augmentation naturally supports hand-related synthesis because the InterAct dataset provides reasonable synthetic hand poses with their correction. We use these poses as references, enabling our augmentation algorithm to generate synthesized hands.
>
>
> We thank the reviewer for the constructive feedback. If the reviewer has any further questions or suggestions, we are happy to discuss them.
>
>
>
>
> [1]  Peng, Xiaogang, et al. Hoi-diff: Text-driven synthesis of 3d human-object interactions using diffusion models. CVPRW, 2025.
>
> [2] Li, Jiaman,et al. Object motion guided human motion synthesis. SIGGRAPH Asia, 2023.
>
> [3] Zeng, Ling-An, et al. Chainhoi: Joint-based kinematic chain modeling for human-object interaction generation. CVPR, 2025.
>
> [4] Li, Jiaman, et al. Controllable human-object interaction synthesis. ECCV, 2024.
>
> [5] Xu, Sirui, et al. Interact: Advancing large-scale versatile 3d human-object interaction generation. CVPR, 2025.
>
> [6] Petrovich, Mathis, et al. Tmr: Text-to-motion retrieval using contrastive 3d human motion synthesis. ICCV, 2023.
>
> [7] Lu, Shunlin, et al. Humantomato: Text-aligned whole-body motion generation. ICML, 2024.
>
> [8] Wu, Zhen, et al. Human-object interaction from human-level instructions. ICCV, 2025.

---

### Official Review · Reviewer_2cfF · 2025-11-01

**Soundness:** 3
**Presentation:** 3
**Contribution:** 3
**Rating:** 6
**Confidence:** 3

**Summary:**

The paper presents LIGHT, a novel framework for generating 3D human-object interaction (HOI) motions from textual descriptions. At the core of the network, the authors propose a pace-induced guidance that avoid the usage of handcrafted contact priors for good contact quality and realism. Instead, LIGHT is formulated with two paths: a uniform pass denopises all modalities (body, hands, object), while a staged pass that uses the outputs from the first pass to guide the noise components. Additionally, a contact-aware shape-spectrum augmentation strategy is proposed to improve the generalization on unseen objects. Extensive experiments have been conducted to present its superior performance over existing techniques.

**Strengths:**

+ The proposed pace-induced guidance and contact-aware shape-spectrum augmentation are novel. Specifically, (1) The pace-induced guidance is proven to provide a data-driven altenative, which is even more effective than the priors used in previous methods. (2) The augmentation builds a informative invariance directly into the training data, leading to improved generatlization.

+ The method demonstrates clear quantitative and qualitative improvements over strong baselines (HOI-Diff, CHOIS, InterDiff) across multiple datasets (InterAct, BEHAVE, OMOMO) and metrics.

**Weaknesses:**

- The method is currently designed and evaluated for interacting with only one single object. Interactions with multiple and complex objects would be more beneficial.

- Comparisons with zero-shot HOI generation methods, such as InterDreamer and ZeroHSI, may also be useful.

- Minor issues: The inference process requires around 72 seconds, which is higher than HOI-DIff and InterDiff (non-guided baselines).

**Questions:**

Besides the weaknesses listed above, the reviewer may have some additional questions:

+ Could the authors illustrate more about the failure mode, such as types of interactions and objects?

+ How's the physical quality, such as penetration issues, of the generated HOI sequences? It might also be better to evaluate for the penetration issues?

---

> ### Author Response · Authors · 2025-11-25
>
> We thank the reviewer for thoughtful and insightful comments. We address the concerns below:
>
> W1: The method is designed for single object
> - We would like to clarify that our experimental evaluation focuses on single-object settings because the benchmark used [5] primarily contains such data. Nonetheless, the proposed method does not inherently restrict the number of objects it can handle. Specifically, the pace-induced guidance is agnostic to the number of objects: objects are represented as tokens in our Transformer architecture, and extending to multiple objects amounts to incorporating additional object tokens. In addition, the contact-aware shape-spectrum augmentation operates independently per object and can be applied to multi-object sequences in a straightforward manner.
> - We appreciate the reviewer’s suggestion and agree that evaluating multi-object interactions is a valuable direction. Extending our experiments to benchmarks featuring multiple and more complex objects is a natural next step that we are eager to explore.
>
>
> W2: Comparisons with zero-shot HOI generation methods
> - For InterDreamer, we provide qualitative comparisons using the same text prompts shown on their website. The visual results are included in the revised supplementary materials (**additional_examples.mp4**). Compared to InterDreamer, our method shows better object dynamics (object is bouncing on the ground), and better contact quality (the yoga ball unnaturally sticks to the left arm in InterDreamer’s result).
> - ZeroHSI targets a different task setting: it takes a 3D scene and a text prompt as input and uses a video diffusion model followed by an HOI reconstruction module. In contrast, we take a static object with a text prompt and generate HOIs in an end-to-end manner. As their code and data are unavailable, and their metric focuses on video alignment, a direct comparison is not feasible.
>
> W3: Inference efficiency compared with baselines
> - We thank the reviewer for pointing out the inference-time difference. As noted in Appendix L1062-L1066, the additional cost arises from two design choices that enable our method’s improved performance: (1) pace-induced guidance requires three forward passes per diffusion step (Equations 5 and 6 in the main paper), and (2) token separation increases the temporal length by a factor of three. These operations are responsible for the observed overhead rather than any fundamental limitation of our approach.
> - Importantly, this cost is orthogonal to the core method and can be reduced with standard techniques. For example, DDIM-style accelerated sampling with fewer timesteps [1, 2] or distillation into a consistency model [3] can significantly improve efficiency without changing our algorithm. We view these engineering optimizations as promising future work and expect that they would bring our inference speed close to that of HOI-Diff and InterDiff.
>
> Q1: Failure mode
> - We observe that our method is more likely to fail on highly dynamic sequences, a challenge shared by all existing methods. We provide visual results in the revised supplementary materials (**additional_examples.mp4**), including our generation and baseline generation. As shown in this example, our generation produces better object dynamics than baselines, but all the methods fail to generate motion with consistent contact.
>
>
> Q2: Physical plausibility and including evaluations on penetration
> - We would like to clarify that a penetration metric (Pene) was included in our evaluation, as noted in L341-L342. Compared with baselines, our method achieves lower penetration overall, as shown in Table A and supplementary video 00:50-01:28. The remaining minor penetration could be partly attributed to artifacts in ground-truth data.
>
> We thank the reviewer for the constructive feedback. If the reviewer has any further questions or suggestions, we are happy to discuss them.
>
> [1] Tevet, Guy, et al. Human motion diffusion model. arXiv:2209.14916, 2022.
>
> [2] Tevet, Guy, et al. Closd: Closing the loop between simulation and diffusion for multi-task character control. arXiv:2410.03441, 2024.
>
> [3] Dai, Wenxun, et al. Motionlcm: Real-time controllable motion generation via latent consistency model. ECCV, 2024.

---

### Author Response · Authors · 2025-11-25
**Summary of Revisions**

We sincerely thank the reviewers for their time and constructive feedback. In response, we have revised our manuscript, with changes highlighted in blue, and also included more visual examples in supplementary materials. The main revisions are summarized below:
- Additional_examples.mp4 is added in supplementary materials, including more visual comparisons.
- Augmentation.mp4 is added in supplementary materials, including more augmentation data and their comparisons with ground truth.
- Additional R-precision results using a larger batch size of 256 in Tables 1, 2, C, D, E, F, G, and H.
- Link the analysis in Section “Impact of Guidance Intensity and Denoising lagging” to Figure 5.
- Correct the notation typos and provide additional explanation for Algorithm 1.

---

### Meta-Review · Area_Chair_1bWX · 2025-12-29

**Summary:**

**Summary**:
This paper presents LIGHT, a data-driven framework for realistic human-object interaction.
The key innovation lies in Learning Implicit Guidance for Human-object inTeraction, where the diffusion forcing idea is implemented to different modalities, and a new sampling schedule is introduced.
The method achieves better quantitative results on most metrics and shows better realistic human-object interaction results than baselines.

**Main strengths**:
- The proposed pace-induced guidance and augmentation are novel, which improves the generalisability.
- The method demonstrates clear improvements over strong baselines across multiple datasets and metrics.

**Main weaknesses**:
- Experimental details: they have been addressed in the reversed paper and rebuttal.

**Suggested decision**:
The paper received initial scores of 6 (2cfF), 4 (sZdf), 6 (woqV), and 2 (wUBQ), and there are no further discussions. However, from the reviewers' comments, the strengths of the paper are strong, while most weaknesses are not big concerns, and they have been discussed in the original paper or be addressed during the rebuttal. Hence, I recommend the final decision as "accept".

**Reviewer Concerns:**

**Interactions with multiple and complex objects would be more beneficial (2cfF)**: Addressed

**Comparisons with more methods (2cfF, sZdf)**: Only show qualitative comparisons.

**Physical plausibility (2cfF)**: Addressed.

**Fine-grained contact (sZdf)**: The results are generated without any post-processing and show better results than baselines.

**Evaluation metrics (sZdf)**: Addressed.

**Experimental and writing details (woqV, wUBQ)**: Addressed.

**Reviewer Scores:**

The paper initially received scores of 6 (2cfF), 4 (sZdf), 6 (woqV), and 2 (wUBQ).
There are no further comments during the discussion.
However, most concerns have been addressed.
Hence, I believe the reviewer may raise their scores during the discussion.

---

### Decision · Program_Chairs · 2026-01-26

Accept (Poster)